# ZmNF-YA1 Contributes to Maize Thermotolerance by Regulating Heat Shock Response

**DOI:** 10.3390/ijms25116275

**Published:** 2024-06-06

**Authors:** Yaling Yang, Zhaoxia Li, Juren Zhang

**Affiliations:** 1Key Laboratory of Plant Development and Environment Adaptation Biology, Ministry of Education, School of Life Sciences, Shandong University, Qingdao 266237, China; yyling2022@163.com; 2Agronomy College, Qingdao Agricultural University, Qingdao 266109, China; zhaoxia_1019@126.com

**Keywords:** maize, thermotolerance, nuclear factor Y, *ZmNF-YA1*, transcriptional regulation

## Abstract

*Zea mays* (maize) is a staple food, feed, and industrial crop. Heat stress is one of the major stresses affecting maize production and is usually accompanied by other stresses, such as drought. Our previous study identified a heterotrimer complex, ZmNF-YA1-YB16-YC17, in maize. ZmNF-YA1 and ZmNF-YB16 were positive regulators of the drought stress response and were involved in maize root development. In this study, we investigated whether ZmNF-YA1 confers heat stress tolerance in maize. The *nf-ya1* mutant and overexpression lines were used to test the role of ZmNF-YA1 in maize thermotolerance. The *nf-ya1* mutant was more temperature-sensitive than the wild-type (WT), while the *ZmNF-YA1* overexpression lines showed a thermotolerant phenotype. Higher malondialdehyde (MDA) content and reactive oxygen species (ROS) accumulation were observed in the mutant, followed by WT and overexpression lines after heat stress treatment, while an opposite trend was observed for chlorophyll content. RNA-seq was used to analyze transcriptome changes in *nf-ya1* and its wild-type control W22 in response to heat stress. Based on their expression profiles, the heat stress response-related differentially expressed genes (DEGs) in *nf-ya1* compared to WT were grouped into seven clusters via *k*-means clustering. Gene Ontology (GO) enrichment analysis of the DEGs in different clades was performed to elucidate the roles of ZmNF-YA1-mediated transcriptional regulation and their contribution to maize thermotolerance. The loss function of *ZmNF-YA1* led to the failure induction of DEGs in GO terms of protein refolding, protein stabilization, and GO terms for various stress responses. Thus, the contribution of ZmNF-YA1 to protein stabilization, refolding, and regulation of abscisic acid (ABA), ROS, and heat/temperature signaling may be the major reason why *ZmNF-YA1* overexpression enhanced heat tolerance, and the mutant showed a heat-sensitive phenotype.

## 1. Introduction

Heat stress is a major stress factor in maize production and is usually accompanied by other stresses such as drought or salinity [1,2]. Heat is one of the major restraints to maize plant growth, development, and yield. Heat during the reproductive stages affects pollen viability, pollen growth, fertilization, and kernel development, resulting in grain yield loss [3,4,5]. Heat stress is expected to be a limiting factor in maize production owing to global warming. The 10 warmest years in the historical record have all occurred in the past decade (2014–2023), and NASA announced the summer of 2023 to be the hottest on record [6,7]. Plant scientists and breeders are currently facing a time-sensitive challenge in understanding how plants, especially crops, sense and tolerate high acute stress to feed the increasing global population under climate change and population explosion scenarios.

To cope with unfavorable environmental conditions, plants have evolved sophisticated mechanisms to respond to abiotic stressors. When plants are subjected to heat stress, stress-specific transcriptome changes are regulated by various transcription factors [8,9]. Nuclear factor Y (NF-Y, also named as heme activator protein or CCAAT-binding factor) are heterotrimeric transcription factor that are widely conserved among eukaryotes [10,11]. NF-Y comprises three subunits, NF-YA, NF-YB, and NF-YC, and is broadly diverse among plant species. The trimer binds to a CCAAT element on the promoter of its target gene to regulate its expression [12,13,14]. The current model of the NF-Y heterotrimeric complex suggests that NF-YA provides sequence specificity to the complex by binding to the minor groove of DNA, whereas the NF-YB/NF-YC subunits serve as a platform that stabilizes NF-YA binding [15,16]. Unlike in mammalian cells, individual NF-Y families have expanded considerably across plant lineages. *Arabidopsis thaliana* (*Arabidopsis*) NF-YA, NF-YB, and NF-YC proteins have 10, 13, and 13 members [12,13,14], and these families have expanded to 14, 18, and 18 members or 16, 19, and 17 members for every subunit in maize identified by different groups [17,18], allowing for enormous combinatorial and functional diversity [12,14,18]. Diversification of plant NF-Ys and their interacting proteins suggests that different trimer combinations have different molecular functions in plants. Recently, researchers have elucidated the importance of NF-Ys in plant growth, development, and stress responses [12,13]. However, detailed insights into the functional diversification of each NF-Y subunit remain limited.

Although the NF-Y complex has not been biochemically characterized in plants, several subunit genes have been studied using genetic and reverse genetic analyses, particularly in the model plant *Arabidopsis*. Several NF-Ys have been found to be involved in the control of organ development and flowering (*NF-YA1*, *NF-YA4*, *NF-YB2*, *NF-YB3*, *NF-YC3*, *NF-YC4*, and *NF-YC9*) [19,20,21]. Mutant analysis showed that NF-YA genes are functionally redundant and are required for embryogenesis, juvenile-to-adult transition, male gametogenesis, and seed development, and these processes occur at least partly through miR169-mediated post-transcriptional regulation [22,23,24]. NF-YAs contribute to abiotic stress by forming specific complexes, such as NF-YB3 and NF-YC2 interacting with bZIP28 and NF-YA4 in ER stress [25]; NF-YA2-NF-YB3-DPB3-1 in the heat stress response [26]; and NF-YC–RGL2 in integrating GA and ABA signaling [27]. Different combinations of trimers, that is, the diversification of plant NF-Ys and their interactors, may be one of the reasons why NF-Ys participate in most, if not all, plant development, growth, reproduction decisions, and stress responses. Their roles are conserved and complicated in crop plants because the gene families encoding each of the three NF-Y subunits have expanded. As these families of TFs play important roles in development and stress responses, plant scientists have focused on understanding their roles and value in crop engineering breeding. In rice, NF-YC12 is a key multifunctional regulator of seed storage substance accumulation, and the NF-YB1-YC12-bHLH144 complex directly activates *Waxy* to regulate grain quality [28,29]. OsNF-YB9, a rice LEC1-like transcription factor, interacts with an endosperm-specific sucrose synthase protein kinase and functions in seed development, reproductive growth, and development [30,31]. OsNF-YB1 regulates grain filling and endosperm development by interacting with ERF [32]. *OsNF-YC2* and *OsNF-YC4* inhibit flowering in rice under long-day conditions [33,34] *OsNF-YC10*, a seed-preferentially expressed gene, regulates grain width by affecting cell proliferation [35]. *OsNF-YC5* and *NF-YC13* are involved in salt stress response, although their contributions differ [36,37]. *OsNF-YA7* confers drought stress tolerance in rice in an ABA-independent manner [38]. Overexpression of *OsHAP2E* confers resistance to pathogens, salinity, and drought and increases photosynthesis and tiller number [39].

Being a widely cultured crop, the biological roles of maize NF-Y genes remain limited, although NF-Y plays an important role in plant development, response, and adaptation to stress. ZmNF-Y genes have been identified in maize with tissue-specific expression patterns at different developmental stages and responses to various abiotic and biotic stressors [40,41,42]. Great success in maize stress-tolerant breeding was reported by Nelson et al., who found that *ZmNF-YB2* confers drought tolerance and leads to improved corn yield in water-limited acres [42]. ZmNF-YA3 interacts with CONSTANS-like (CO-like) and flowering promoting factor 1, and the complex regulates flowering time and abiotic stress responses [43]. *ZmNF-YC12* was highly induced by drought and rewatering treatments. The Gal4-LexA/UAS system and a transactivation analysis demonstrated that ZmNF-YC12 is a transcriptional activator that regulates drought resistance and recovery ability [17]. Silencing of *ZmNF-YC12* reduced net photosynthesis and other physiological indexes and led to the reduction of drought resistance and the recoverability of maize [17]. ZmNF-YC13 functions as a transcriptional regulator and, together with ZmNF-YBs and ZmNF-YA3, affects plant architecture by regulating the expression of *ZmWRKY76* and *ZmBT2* [44]. The ZmNF-YC1-ZmAPRG pathway modulates low phosphorus tolerance in maize [45]. As plant biologists and breeders, we investigated the role of NF-Ys in abiotic stress response in maize.

Our previous study showed that overexpression of *ZmNF-YB16* improved drought resistance and yield by enhancing photosynthesis and the antioxidant capacity of maize plants [46], after which one heterodimer complex, ZmNF-YB16-YC17, was identified in maize and entered the nucleus to form a heterotrimer with ZmNF-YA1 or ZmNF-YA7 when subjected to osmotic stress [47]. ZmNF-YA1 and its interactor, ZmNF-YB16, are positive regulators of abiotic stress responses and are involved in the root development of maize [47]; however, the downstream networks of the two do not completely overlap. To be a positive regulator of osmotic stress and to work together with ZmNF-YB16-YC17, we were interested in the role of ZmNF-YA1 in the heat stress response of maize and whether it could be a candidate for genetic engineering breeding. In this paper, the morphology, growth of maize under moderate heat stress conditions were evaluated using *ZmNF-YA1* transgenic lines and mutant. Similar to the drought tolerance observed previously, *ZmNF-YA1* was found to be a positive regulator of heat tolerance. Higher malondialdehyde content and ROS accumulation were observed in the mutant, followed by WT and overexpression lines after heat stress treatment, while an opposite trend was observed for chlorophyll content. Transcriptome analysis showed that the loss function of *ZmNF-YA1* led to broad biological changes from transcriptional regulation to photosynthesis. One of the processes regulated by ZmNF-YA1 and played important role in heat stress responses is protein refolding. The contribution of ZmNF-YA1 to protein stabilization and the regulation of abscisic acid (ABA), ROS, and heat/temperature signaling may be the major reason why *ZmNF-YA1* overexpression enhanced heat tolerance and the mutant showed a heat-sensitive phenotype.

## 2. Results

### 2.1. Expression Profile of ZmNF-YA1 in Maize Heat Stress Response

Our previous result showed that ZmNF-YA1 could interact with the ZmNF-YB16-YC17 heterodimer to exert its biological function [47]. The evolutionary tree was conducted using ZmNF-YA1 and the members in this subgroup. Sequence alignment showed that five maize NF-YA members existed in the clade of ZmNF-YA1, with a high sequence similarity to *Arabidopsis* NF-YA3 (AtNF-YA3) and OsNF-YA1 in rice (Appendix A). *ZmNF-YA1* expression was the highest in embryos, followed by ears and leaves (Appendix A). The expression of *ZmNF-YA1* was higher in leaves than in roots during the vegetative growth stage [47].

Changes in the expression of *ZmNF-YA1* in maize during heat stress were examined using quantitative reverse transcription real-time PCR (qRT-PCR), as shown in Appendix A. Leaves from 10-day-old seedlings subjected to different heat stress treatments were used in this study. Three different temperatures, 39 °C (moderate heat stress), 42 °C (heat shock stress), and 45 °C (severe heat stress), were selected for the heat stress treatment. As shown in Appendix A, *ZmNF-YA1* was an early heat-induced gene, as upregulation was observed after 15 min of heat treatment, with the highest peaks at 1 h, followed by a decrease. As the temperature increased, a higher induction of *ZmNF-YA1* and time to reach the highest levels were observed. Expression pattern analysis indicated that ZmNF-YA1 may participate in maize heat stress response.

### 2.2. Expression of ZmNF-YA1 in the Mutant and Overexpression Lines

Mutant and overexpression lines were used to test the effects of *ZmNF-YA1* on maize thermotolerance. The *nf-ya1* homozygous mutant lines, which had no *ZmNF-YA1* transcripts, and its wild-type control maize inbred line W22 were used in this study. Another group used was T4 generation homozygous *ZmNF-YA1* overexpression lines and their transgene donor line DH4866. Independent *ZmNF-YA1* overexpression lines were produced by *Agrobacterium tumefaciens*-mediated maize shoot-tip transformation described in our previous study [47]. Mutant and transgenic maize plants were confirmed using a PCR assay with *bar* gene-specific primers, and homozygous lines were selected for further analysis. The expression levels of *ZmNF-YA1* were obviously higher than those in DH4866 under heat stress treatment because the transgene *ZmNF-YA1* was driven by the *AtRD29A* promoter [48], which is an abiotic stress-inducible promoter in *Arabidopsis*, in the overexpression lines. Major differences were detected between transgenic, WT, and mutant plants under normal control and heat stress conditions. Before treatment, *ZmNF-YA1* transcript levels in the overexpression lines were 1.75- to 2.04-fold higher than those in the WT, whereas in the mutant, they were approximately 0.1-fold higher than those in the W22. When subjected to heat stress, *ZmNF-YA1* transcript levels were induced by approximately 3.5-fold in the WT (DH4866 and W22) plants after 1 h of heat stress treatment compared to the value before heat treatment. *ZmNF-YA1* transcript levels were much higher (Figure 1) because of the heat-induced transgene (driven by the *AtRD29A* promoter) and endogenous *ZmNF-YA1* in the overexpression lines.

After the preliminary test, a time-course experiment was performed at three different temperatures: moderate heat stress (39 °C), heat shock stress (42 °C), and severe heat stress (45 °C). As shown in Figure 2, *ZmNF-YA1* was upregulated after 15 min of heat treatment in all lines, except *nf-ya1*. The highest peaks were observed after 1 h of treatment at 39 °C moderate heat stress and 42 °C heat shock stress, whereas the highest peak was observed after 0.5 h of treatment at 45 °C severe heat stress. The *ZmNF-YA1* overexpression lines had higher *ZmNF-YA1* transcripts, and the mutant had lower *ZmNF-YA1* transcripts than the WT before and after heat stress.

### 2.3. ZmNF-YA1 Functions as a Positive Regulator in Maize Thermotolerance

We investigated whether *ZmNF-YA1* confers heat stress tolerance in maize because a strong induction of *ZmNF-YA1* was observed in response to heat stress. To answer this question, the phenotypes of the transgene donor inbred line DH4866, *ZmNF-YA1* overexpression plants, mutant, and its control W22 in response to heat stress were examined. A 45 °C severe heat shock stress (Figure 3) was used to determine the survival rate and damage of plants.

Under normal conditions, all plants showed vigorous growth, although the aerial parts of the transgenic lines were slightly larger than those of the WT plants. Major differences were evident after 6 h of heat treatment at 45 °C (Figure 3B). Wilting was more severe in the mutants than in the wild-type and transgenic lines. The survival rate showed similar trends, that is, a positive correlation between *ZmNF-YA1* levels and heat tolerance was observed. Malondialdehyde (MDA) and reactive oxygen species (ROS) are indicators of cell damage. Higher MDA content and ROS accumulation were observed in the mutant line, followed by WT and overexpression lines after heat stress treatment (Figure 3E,F and Appendix A), whereas an opposite trend was observed for the chlorophyll content. The MDA content of the mutant line was approximately 140% that of WT at 6 h heat stress treatment, whereas the MDA content of the overexpression lines was approximately 80% that of WT (Figure 3C). Chlorophyll a and chlorophyll b contents were reduced after 6 h of severe heat stress, particularly chlorophyll b. All differences in morphological and physiological changes demonstrated that the *nf-ya1* plants were heat-sensitive, and overexpression of *ZmNF-YA1* enhanced maize heat tolerance. *ZmNF-YA1* positively regulated maize thermotolerance.

### 2.4. Comparative Transcriptome Analysis of Maize nf-ya1 and W22 in Heat Stress Response

To understand the role of *ZmNF-YA1* in maize thermotolerance, RNA-seq was used to analyze transcriptome changes in W22 and *nf-ya1* plants in response to heat stress (GES268429). Differentially expressed genes (DEGs) were identified by comparing the genotypes (*nf-ya1* vs. W22) and growth conditions (treatment vs. normal) (Appendix A and Appendix A). Principal component analysis (PCA) showed obvious differences between the *nf-ya1* mutant and W22 under normal conditions, as well as under heat stress treatment (Appendix A). Based on a generalized linear model analysis using the Dr. Tom system from BGI (https://www.bgi.com/global/service/dr-tom, accessed on 23 April 2023), DEGs from the comparison of time-course and between genotypes were identified with an adjusted FDR less than 0.01 and an absolute fold value greater than 1 (Appendix A). Under normal conditions, 4147 DEGs showed higher expression levels in the *nf-ya1* mutant than in W22, and 2014 DEGs showed lower expression levels in the *nf-ya1* mutant than in W22 (Appendix A). When subjected to heat stress treatment for 1 h, 4253 DEGs had higher expression levels in the *nf-ya1* mutant than in W22, 2044 DEGs showed lower expression levels in the *nf-ya1* mutant than in W22, 2446 DEGs had higher expression levels in the *nf-ya1* mutant than in W22, and 1246 DEGs had lower expression levels in the *nf-ya1* mutant than in W22 when recovered (Appendix A). Under all conditions, more upregulated than downregulated DEGs were observed in the comparison of *nf-ya1*/W22 (Appendix A, right panel). To confirm these results, 10 genes with different transcript abundances were validated using qRT-PCR (Appendix A). The expression levels of these genes were consistent between the two detection methods.

Based on the gene expression profiles of the different genotypes and heat stress conditions, the DEGs were divided into different groups, as shown in Figure 4A,B. The heat stress response DEGs in *nf-ya1* and W22 were grouped into seven clusters via *k*-means clustering (Figure 4C), and the Akaike information criterion was used to select the optimal number of clusters K. The representative expression profiles of *nf-ya1* and W22 in response to heat are shown in Figure 4D. Among the seven clusters, Clusters 1, 2, 6, and 7 were the most interesting because they were ZmNF-YA1-regulated DEGs and involved in the heat stress response. The DEGs in Cluster 1 showed a heat stress-induced pattern in *nf-ya1* and W22; however, the induced fold changes were much higher in W22 than in *nf-ya1* under heat stress conditions. The DEGs in Cluster 2 were heat stress-induced; however, the induced fold changes were much lower in W22 than in *nf-ya1* under heats tress conditions. These DEGs were heat-induced genes that were repressed (Cluster 1) or activated (Cluster 2) by ZmNF-YA1. In contrast, Clusters 6 and 7 were down-regulated by heat and were positively (Cluster 7) or negatively (Cluster 6) regulated by ZmNF-YA1. In addition to the DEGs regulated by heat and ZmNF-YA1, thousands of DEGs were found when comparing *nf-ya1* and W22. GO and pathway analysis no major changes were observed in the heat stress treatment between genotypes, however the more stress related processes in *nf-ya1* compared to W22 (Appendix A). implying that ZmNF-YA1 regulates multiple biological processes, including heat stress response.

### 2.5. ZmNF-YA1 Contributes to the Transcriptional Regulation in Maize Heat Stress Response

To examine the role of ZmNF-YA1, Gene Ontology (GO) enrichment analysis of the DEGs in *nf-ya1* compared to W22 was performed (Figure 5, Appendix A). GO analysis of genes differentially expressed in *nf-ya1* compared to W22 at both normal and heat stress conditions showed multiple biological processes were significantly adjusted in the *nf-ya1,* such as carbon utilization, growth, development, and response to stimulus (Appendix A). Interestingly, the GO terms in response to stimulus in the biological function and the terms protein folding chaperone in the molecular function were significantly altered in the *nf-ya1* compared to W22. Pathway enrichment analysis showed a similar result as that the protein processing in the endoplasmic reticulum (ER) was significantly enriched (Appendix A). When mapping to the pathway, both the molecular chaperones in Unfolded Protein Response (UPR) and the molecular chaperones in the cytoplasmic were regulated by *ZmNF-YA1*, such as calreticulin (CRT); luminal binding protein (BIP and GRP94) in the ER; and the heat shock protein (HSP) 40, HSP70, and HSP90 in the cytoplasmic (Appendix A).

Based on the DEGs between the different genotypes and heat stress conditions, GO enrichment analysis of the different clade DEGs was performed too. In Cade 1, which was induced by ZmNF-YA1 (lower expression level in *nf-ya1* compared to W22) and heat stress, the GO terms of protein refolding, protein stabilization, chaperone-mediated protein folding, protein complex oligomerization, de novo post-translational protein folding, and protein folding were considerably enriched (Figure 5A). GO terms in stress responses, such as response to H_2_O_2_, response to heat, response to hypoxia, cellular response to decreased oxygen levels, response to reactive species, response to oxygen levels, abscisic acid-activated signaling pathway, cellular response to abscisic acid stimulus, response to salt stress, response to temperature stimulus, and response to osmotic stress, were significantly enriched in this clade too.

In contrast to the DEGs induced by ZmNF-YA1 and heat stress, the GO terms enriched in the clades of DEGs repressed by ZmNF-YA1 and heat stress were different. As shown in Figure 5B, the GO terms of organ development and growth were considerably enriched, particularly in the reproductive development processes. The GO terms photomorphogenesis, floral organ development, flower development, and reproductive shoot system development were at the top, followed by several biosynthesis and metabolic processes related to growth, such as cell wall organization and biogenesis. Stress causes growth retardation, and developmental stage transition was observed in the plant heat stress response. These clades of DEGs (ZmNF-YA1 and heat stress) might contribute to growth and development under heat stress. Various GO terms were found in the clades shown in Figure 5C,D, indicating that ZmNF-YA1 regulates multiple processes, including the heat stress response.

### 2.6. Effect of ZmNF-YA1 on Maize’s Heat Stress Response

Heat shock transcription factors are transcription factors that play a central role in activating the expression of heat shock proteins, the molecular chaperones in the cell [51,52]. As shown in Figure 6A and Appendix A, heat shock transcription factor (*hsftf*)*10* and *hsftf13* were induced by moderate heat stress and recovered to normal levels; however, their expression levels were lower in *nf-ya1* after 1 h of heat stress treatment for *hsftf10* and *hsftf13*. Their expression levels in the *ZmNF-YA1* overexpression plants were higher WT. In contrast, 12 of the identified heat shock transcription factors were induced by moderate heat stress and then recovered to normal levels, whereas in *nf-ya1*, the levels of these 12 heat shock transcription factors were considerably higher than those in the W22 plants (Figure 6B and Appendix A). The loss of function of *ZmNF-YA1* not only changed the expression of heat shock transcription factors when subjected to heat stress but also led to changes under normal conditions and during the recovery stage (Figure 6B and Appendix A). For example, *hsftf10* and *hsftf18* showed major differences in the expression levels between the *nf-ya1* line and W22 before and after heat stress, with a higher level in W22, which faded at R2h.

Heat shock proteins function as chaperones to prevent protein misfolding and aggregation and were grouped into different classes according to molecular weight: small heat-shock proteins (sHSPs), heat shock protein (HSP)40, HSP60, HSP70, HSP90, and HSP100 [53,54,55,56]. The 31 sHSPs were upregulated by heat in our experiments. It is worth noting that the loss of ZmNF-YA1 function led to a lower expression of sHSPs under normal and heat stress conditions (Figure 6B, left panel and Appendix A). DnaJ encodes the DnaJ protein, which is also known as HSP40. The expression of 15 DnaJ genes was lower in the mutant line than in the W22 line. Similar to sHSPs, heat-induced expression of HSP70s and HSP90s was observed, and some were expressed at lower levels in *nf-ya1* (Figure 6C).

### 2.7. Validation of the NF-YA1 Function in the Expression of Candidate Target Genes

To explore the downstream genes of ZmNF-YA1, we used PlantPAN3.0 (http://plantpan.itps.ncku.edu.tw, accessed on 20 September 2023) for cis element searches of the identified DEGs. NF-Y complex recognizes and binds with high affinity and sequence specificity to the cis-acting element CCAAT motif [14]. The number and localization of the predicted NF-YA binding core “CCAAT box” in the promoter regions of the DEGs were analyzed. Promoters from six candidate ZmNF-YA1 target genes were selected for a promoter binding assay in the maize protoplast transient system (Figure 7A,B). Approximately 2000 bp promoter sequences from two plant hormone-related genes, *bZIP-transcription factor 100* (*bZIP100*, Zm00001eb373300) and *jasmonoyl-l-isoleucine hydrolase 5* (*JIH5*, Zm00001eb211100), four stress-related protein coding genes *asparagine synthetase 3* (*ASN3*, Zm00001eb013430), *hypoxia response unknown protein 26* (*HUP26*, Zm00001eb406490), *general regulatory factor 1* (*GRF1*, Zm00001eb080570), and *NOD26-like membrane intrinsic protein 2c* (*NIP2c*, Zm00001eb372380) were amplified and inserted into the Dual-luciferase (LUC) vector to drive the expression of *LUC*. As shown in Figure 7A, all promoters harbored multiple CCAAT cis elements that were recognized and bound by the NF-Ys. The CDS region of *ZmNF-YA1* was inserted into the plant transient expression vector to simulate the overexpression of *ZmNF-YA1* in plant cells (Figure 7B). The promoter activity in different lines (W22, *ZmNF-YA1* overexpression, and mutant cells) under normal and heat treatments was analyzed and indicated by the value of LUC/REN (Figure 7C). Among the six candidates selected, ZmNF-YA1 functioned as a positive regulator to bind and activate the expression of *ASN3*, *HUP26*, *bZIP100*, and *GRF1*. When the expression levels of *ZmNF-YA1* were enhanced, a higher LUC/REN and more transcripts were detected in the promoter binding assay and transcriptome analysis. In addition, co-expression patterns were observed for these candidates with *ZmNF-YA1*, all of which showed heat stress-induced expression and a positive correlation with the levels of *ZmNF-YA1* (Figure 7C and Appendix A). The expression of *JIH5* and *NIP2c* negatively correlated with the level of *ZmNF-YA1*, and when *ZmNF-YA1* was overexpressed or induced, the expression levels of *JIH5* and *NIP2c* were reduced. ZmNF-YA1 could work as a transcriptional repressor, i.e., could bind to the promoter and downregulate the expression of *JIH5* and *NIP2c*.

## 3. Discussion

### 3.1. ZmNF-YA1 Is a Genetic Manipulation Target for Improving Thermotolerance and Drought Tolerance

The NF-YA gene family comprises a class of conserved transcription factors that are key regulatory factors in plant development and abiotic stress responses. Drought and high temperature are the two major abiotic stresses that reduce maize production and usually occur simultaneously. In our previous study, we identified a ZmNF-YA1-YB16-YC17 heterotrimer. Two components of this complex, ZmNF-YA1 and ZmNF-YB16, were found to be positive regulators of osmotic/drought stress [46,47]. Overexpression of *ZmNF-YA1* or *ZmNF-YB16* in maize considerably improves drought resistance and yield [46,47]. The expression level of *ZmNF-YA1* was induced by 12% PEG6000, 120 mM NaCl, and 42 °C stress treatment, although with varying degrees, suggesting that ZmNF-YA1 could be a regulator in various abiotic stress responses in maize. We were interested in the role of ZmNF-YA1 in maize heat stress response and aimed to determine whether it could be used as a genetic manipulation target for maize heat and drought tolerance. Expression profile analyses showed that as the temperature increased, a higher induction of *ZmNF-YA1* and time to reach the highest levels were observed. This indicated that ZmNF-YA1 may participate in maize heat stress response. Different heat stress treatments were used to evaluate the thermotolerance of the *nf-ya1* and *ZmNF-YA1* overexpression lines and their WT. Under heat stress conditions, transgenic maize plants with increased *ZmNF-YA1* expression showed heat tolerance, and *nf-ya1* plants showed a heat-sensitive phenotype based on the responses of several heat stress-related parameters, including chlorophyll content, MDA content, ROS accumulation, leaf wilting, and the expression of heat shock molecular markers. All differences in morphological and physiological changes demonstrated that the *nf-ya1* plants were heat-sensitive, and the overexpression of *ZmNF-YA1* enhanced maize heat tolerance. ZmNF-YA1 positively regulates maize thermotolerance. Although the mechanism of maize adaptation to heat or drought stress is unclear, ZmNF-YA1 or its ZmNF-YA1-YB16-YC17 heterotrimer could be a hub shared by multiple abiotic stress responses, particularly heat and drought. Taken together, these results suggest that ZmNF-YA1 is an ideal genetic manipulation target for improving maize thermotolerance and drought tolerance.

Sequence alignment showed that five maize NF-YA members and three rice NF-YA members existed in the clade of ZmNF-YA1, with a high sequence similarity to AtNF-YA3. AtNF-YA3 and NF-YA8 were functionally redundant and required in early embryogenesis in *Arabidopsis* [22]. Overexpression of *AtNF-YA3,* or its orthologs *Glycin max (Gm)NF-YA3,* enhanced drought tolerance in *Arabidopsis* [53]. Individual AtNF-YA3 has expanded to three members in rice (OsNF-YA1, OsNF-YA2, and OsNF-YA6) and a small five members subgroup in maize, allowing for their functional diversity such as in stress responses. Overexpression of an *OsNF-YA2*/*OsHAP2E* enhanced resistance to salinity and drought and increases photosynthesis and tiller number in rice [39]. In maize, the expression level of *ZmNF-YA1* was the highest in this subgroup. Moreover, *ZmNF-YA1* was heat-induced, while the other four in this clade were somehow heat-inhibited (Appendix A). In our previous study, when screening the potential ZmNF-YAs to ZmNF-YB16/YC17, ZmNF-YA1, and ZmNF-YA7 were the only two that could interact with ZmNF-YB16/YC17 to form the heterotrimer [47]. ZmNF-YA1 could be the key orthologous in maize in abiotic stress responses, especially heat and drought stress.

### 3.2. Failure to Launch Full-Fledged Heat Stress Response in nf-ya1 in Response to Heat Stress

When plants are subjected to heat stress, one of the symbolic events is the dramatic induction of heat stress response genes, known as heat stress response. The heat stress response is homeostatic mechanisms that mitigate the damage from heat stress and protect plants from further stress. In transcriptional regulation in response to heat stress, heat shock transcription factors play a central role by activating the expression of constellated HSPs, which are then involved in plant heat stress response and thermotolerance [51,52]. Plant heat shock transcription factors are divided into three conserved evolutionary classes (A, B, and C) according to the structural features of their oligomerization domains. Class A heat shock transcription factors are essential for transcriptional activation, which triggers the immediate expression of downstream genes [57,58]. The roles of the regulation mechanisms of class A heat shock transcription factors have been well studied in Arabidopsis model plants, such as HSFA1 transactivation activity regulation by interaction with HSP70 and HSP90 [59,60] and reversible phase separation [61]; HSFA2 as a hub in heat stress-induced transcriptional regulation [62,63]; and the relationship between HSFA3, DREB2s, and ABA-signaling with HSFA6b [64]. However, overexpression of a single heat shock transcription factor or heat shock protein coding gene had modest or little effect on thermotolerance, suggesting that heat shock transcription factors and heat shock proteins act synergistically to confer thermotolerance. Class B and C heat shock transcription factors do not have AHA motifs and activator functions of their own or function as regulators of transcriptional heterodimers [57,58].

In this study, the loss of function of *ZmNF-YA1* affected the expression of several heat shock transcription factors and their response to heat stress. Maize heat shock transcription factors coding genes *hsftf10* and *hsftf13* were induced by moderate heat stress and then recovered to normal levels. However, their expression levels were lower in *nf-ya1* after 1 h of heat stress treatment. Moreover, 12 of the identified heat shock transcription factors were induced by moderate heat stress and then recovered to normal levels, whereas, in *nf-ya1*, the levels of these 12 heat shock transcription factors were considerably higher than those in W22 plants (Figure 6B and Appendix A). The loss of function of *ZmNF-YA1* not only changed the expression of heat shock transcription factors when subjected to heat stress but also led to changes under normal conditions and during the recovery stage. Heat shock transcription factors-mediated transcriptional regulation has been noted as a critical role hub for plants against unfavorite temperatures. The abolishment of *ZmNF-YA1* led to the expression changes of 23 heat shock transcription factors, although the trends or ranges varied. However, promoter analysis of some of the heat shock transcription factors showed they did not have the canonical NF-YA binding “CCAAT box”. It is interesting and worth pondering why so many heat shock transcription factors coding genes were simultaneously affected by altering the expression of *ZmNF-YA1*. One possible reason could be some unknown cis-elements existed in their promoters that ZmNF-YA1 could recognize and bind, and hereafter regulated their expression. Another possible reason could be the alternation of the cell homeostasis when *ZmNF-YA1* was abolished. In the study, some stress response genes and pathways were altered under normal conditions, such as the protein processing system. This suggests that the differentially expressed heat shock transcription factors could be the feedback regulation of the altered cell vigilance.

HSPs function as chaperones to prevent protein misfolding and aggregation and were grouped into different classes according to molecular weight: sHSPs, HSP40, HSP60, HSP70, HSP90, and HSP100 [54,55,56]. The sHSPs are often the first line of defense in response to stresses, including heat. When we examined the expression of *HSPs*, the most important downstream targets of heat shock transcription factors, we found that the loss function of *ZmNF-YA1* resulted in lower expression of *sHSPs* under normal and heat conditions, especially the sHSPs and DnaJ coding genes, which are recognized as the “paramedics of the cell” [56] when subjected to unfavorable environment conditions. These results suggest that ZmNF-YA1 is involved in regulating the expression of multiple heat shock transcription factors and contributes to the transcriptional regulation of maize heat stress response. Failure to launch a full-fledged heat stress response in *nf-ya1* could be the major reason why *nf-ya1* showed a heat-sensitive phenotype and why *ZmNF-YA1* overexpression enhanced plant thermotolerance.

### 3.3. ZmNF-YA1 Promotes Protein Refolding and Stability When Subjected to Heat Stress Treatment

To understand the role of ZmNF-YA1 in maize thermotolerance, RNA-seq was used to analyze transcriptome changes in W22 and *nf-ya1* plants in response to heat stress. Based on the expression profiles, heat stress response-related DEGs in *nf-ya1* and W22 were grouped into seven clusters via *k*-means clustering. Among the seven clusters, Clusters 1, 2, 6, and 7 were the most interesting because they were ZmNF-YA1-regulated DEGs and were also involved in heat stress response. GO enrichment analysis of DEGs from different clades DEGs was performed. In clade 1, which was induced by ZmNF-YA1 and heat stress, the GO terms of protein refolding, protein stabilization, chaperone-mediated protein folding, protein complex oligomerization, de novo post-translational protein folding, and protein folding were considerably enriched (Figure 5A). Most of these were HSP-coding genes that were regulated by heat shock transcription factors (Figure 6). HSPs are the most important chaperones in the folding/refolding systems that alleviate protein misfolding and aggregation. One of the causes of cell injury is protein misfolding, protein aggregation, and damage to enzyme catalytic domains. The major impacts of heat stress at the cellular level are alteration of membrane fluidity, which disrupts photosynthesis and respiration, misfolding of proteins, accumulation of protein aggregates that cause proteotoxic stress, production of ROS to deleterious levels, metabolic imbalance, and cytoskeleton dismantling [65,66,67]. Protein folding/refolding in the cytoplasm and endoplasmic reticulum helps mitigate damage from heat stress and protects plants from further stress [65,66,67]. The contribution of ZmNF-YA1 to protein stabilization, refolding, and regulation of ABA, ROS, and heat/temperature signaling may be a major reason for the enhanced heat tolerance of the *ZmNF-YA1* overexpression lines, and the mutant showed a heat-sensitive phenotype. An interesting link between ZmNF-YA1-heat stress response and the protein stabilization system was found to contribute to maize thermotolerance. In conclusion, ZmNF-YA1 is beneficial to protein refolding and stabilization when subjected to heat stress.

ZmNF-YA1 is a positive regulator in maize thermotolerance and drought tolerance. We were interested in the shared and stress-specific of the ZmNF-YA1-mediated transcriptional regulation in response to heat and drought stress. The heat shock transcription factors and the mediated heat stress response were specifically found in heat stress except for some of them, which were also involved in drought stress response. More DEGs regulated by ZmNF-YA1 and heat stress were identified in this study when compared to drought stress treatment, although a moderate 39 °C was used. The ZmNF-YA1 regulated ABA-dependent and independent signaling pathways, and other abiotic stress responses GO/pathways were shared in response to both heat and drought stress. The cell damage caused by high temperature was more severe than the osmotic stress. Different from heat shock transcription factors, HSPs, especially the sHSPs were differentially expressed in response to drought. The “paramedics of the cell” not only work in heat stress but also osmotic stress, although they are first identified in heat stress treatment. The alternation of the cell homeostasis by abolishing *ZmNF-YA1* could be one of the major reasons for the changed heat and drought stress tolerance, with a positive correlation with the *ZmNF-YA1* level and tolerance. The ZmNF-YA1 regulated GO/pathways in various stresses contribute to the enhanced basal resistance to multiple stresses such as heat and drought. In addition, ZmNF-YA1 also participates in the stress-specific response in a stress-induced manner. The present study reveals the function of ZmNF-YA1 in maize heat and drought tolerance and provides an interesting hub for understanding the molecular mechanisms in maize response to abiotic stresses.

## 4. Materials and Methods

### 4.1. Plant Materials

Maize *nf-ya1* homozygous mutant and *ZmNF-YA1* homozygous independent overexpression lines obtained by Yang et al. [47] were used in this study. Briefly, maize *nf-ya1* mutants carrying a *Mu* insertion in Zm00001eb005690 were obtained from the UniformMu population UFMu-06468 in an inbred line W22 background and produced by self-pollination for three generations. The mutant had no *ZmNF-YA1* transcripts. *ZmNF-YA1* overexpression plants produced by *Agrobacterium tumefaciens*-mediated shoot-tip transformation were obtained from the elite inbred maize line DH4866. *nf-ya1* homozygous mutant and T4 generation homozygous *ZmNF-YA1* overexpression lines were used for phenotype and heat stress treatment analysis. Seedlings of the inbred line DH4866 were used for gene expression profile analysis.

### 4.2. Plant Growth and Stress Treatment

Plant morphology was analyzed primarily during the seedling stage. Briefly, maize seeds from different lines were sown in pots (12 cm diameter × 15 cm height) containing homogeneous loam. The plants were grown under a 26 °C/22 °C (day/night) temperature regimen at a photon flux density of 700 μmol m^−2^ s^−1^ with a 16 h/8 h light/dark cycle in a chamber with approximately 65% relative humidity. After germination, the seedlings were watered under normal conditions. During heat stress treatment, half of the plants were transferred to a chamber at a set temperature. Severe heat stress at 45 °C was applied to test the thermotolerance of *ZmNF-YA1* independent overexpression lines and their control line (DH4866, donor inbred line), as well as *nf-ya1* mutant and its wild-type control W22. Plants were photographed, and leaves were collected before and after 6 h of heat stress treatment and 2 d after recovery (0, 6 h, and R2d) for a number of heat stress-related parameters, including chlorophyll content, MDA content, ROS accumulation, leaf wilting, and expression of heat shock molecular markers. Approximately 0.1 g of the leaf lamina (excluding the midrib) from the middle of the first fully expanded leaf (from top, at the three-leaf stage) was used for physiological parameter measurements, and at least three biological replicates were used. Each biological replicate contained the leaf fragments from 3–4 plants grown under the same conditions. For gene expression profile analysis, plants from different lines were germinated and watered normally; then, 10-day-old maize seedlings were subjected to 39 °C, 42 °C, and 45 °C heat stress for 10 h, respectively, and then recovered. Leaf samples from three biological replicates were collected at different time points during heat stress (0, 0.25, 0.5, 1, 2, 4, 6, 8, and 10 h during heat stress) and 2 h after recovery (R2h) for qRT-PCR analysis.

### 4.3. RNA Extraction, qRT-PCR Analysis, and RNA-seq Analysis

RNA was extracted from a small sample (~0.1 g) of leaf from the middle of the first fully expanded leaf. The sample was ground to a powder in liquid nitrogen, RNA was isolated using TRIzol reagent (Sangon Biotech, Shanghai, China), and cDNA synthesis with the RT reagent kit was performed according to the manufacturer’s protocol (DRR420, TaKaRa, Dalian, China). The expression patterns were detected using qRT-PCR. qRT-PCR was conducted with the SYBR^®^ RT-PCR Kit (RR037Q, TaKaRa, Dalian, China) using 10 μL reactions for each sample and amplified through 40 cycles on an ABI7500 (Thermo Fisher Scientific, Foster City, CA, USA). Maize 18S rRNA was used as the internal control [49]. Primers were designed using NCBI/primer-BLAST and are listed in Appendix A. Relative gene expression levels were calculated using the 2^−ΔΔCt^ method [50]. Each experiment was repeated thrice, and three biological replicates were used.

The BGISEQ-500 platform was used for RNA-seq analysis. RNA quality and quantity were examined using UV absorbance spectroscopy (Nanodrop 2000, Thermo Fisher Scientific, Foster City, CA, USA), gel electrophoresis, and an Agilent 2000 BioAnalyzer (Agilent, Technologies, Santa Clara, CA, USA). Library construction, sequencing, and primary bioinformatics analyses were performed by Beijing Genomic Institute (BGI) Tech Solutions Co., Ltd. (Beijing, China), according to the standard procedure. Bioinformatic analysis for digital gene expression profiling was performed according to the bioinformatics analysis procedure of BGI Tech. After quality control raw reads were filtered to obtain clean reads aligned to the reference sequences (B73 V5). Alignment data were used to calculate the distribution of reads on the reference genes and mapping ratio. The criteria of FDR ≤ 0.01 and the absolute value of log_2_Ratio ≥ 1 were used as thresholds to determine the significance of differences in gene expression. GO analysis was performed using AgriGO (http://bioinfo.cau.edu.cn/agriGO/, accessed on 20 May 2023) and GENEONTOLOGY (https://geneontology.org, accessed on 25 May 2023). Pathway analyses were performed using the Kyoto Encyclopedia of KEGG (http://www.genome.jp/kegg/, accessed on 28 June 2023). The R software (R4.3.2) and its packages were used for cluster analysis, heatmap plotting, boxplot plotting, and statistical analysis (https://www.r-project.org, accessed on 15 December 2023).

### 4.4. Chlorophyll and MDA Content Analysis, as Well as Cell Staining

Chlorophyll content was estimated using Aron’s method [68] with minor modifications. Approximately 0.1 g leaves samples were collected, and chlorophyll was extracted using 5 mL of 80% aqueous acetone for two days in the dark, until the leaves were completely whitened. The concentrations of chlorophyll a and b were determined by measuring the absorption at 663, 645, and 470 nm using a UV spectrophotometer. The amount of chlorophyll present in the extract (mg chlorophyll per gram of tissue) was calculated using the following equations: chlorophyll a = (12.72A663–2.59A645)V/(1000 m) and chlorophyll b = (22.88A645–4.67A663)V/(1000 m). The results are expressed as mg/g, where A = absorbance at a specific wavelength, V = final volume of chlorophyll extract in 80% acetone, and W = weight of fresh tissue extracted. To measure the MDA content, leaves (~0.1 g) were homogenized in 2 mL of a chilled reagent composed of 0.25% (*w*/*v*) thiobarbituric acid in 10% (*w*/*v*) trichloroacetic acid and then centrifuged at 10,000 rpm for 20 min. The supernatant was heated at 95 °C for 30 min, quickly cooled on ice, and centrifuged at 10,000 rpm for 20 min. Absorbance at 532 nm (A532), 600 nm (A600), and 450 nm (A450) was measured using a UV spectrophotometer. The MDA content was calculated as: MDA content = 6.45 × (A532 − A600) − 0.56 × A450 [69]. The middle part of the second fully extended maize leaf was used for ROS staining, as previously reported by [70]. Fully expanded leaves (approximately 1 cm) were sampled, washed with distilled water, and placed in 2 mL buffer solution containing nitroblue tetrazolium (NBT, 6 mM solution prepared in sodium citrate (pH 6.0)), 3,3-diaminobenzidine (DAB, 1 mg/mL, prepared in double distilled water (pH 3.8)), or Evans Blue (0.5%). The dipped samples were vacuum-infiltrated for 10 min at 60 KPa pressure and then incubated at room temperature overnight. After incubation, the samples were dipped in absolute ethanol and then kept in a water bath (100 °C) until the chlorophyll was completely removed from the samples, which were then cooled and dipped in 20% glycerol, and images were captured.

### 4.5. Sequence Analysis

Sequences were downloaded from the MaizeGDB (www.maizegdb.org, accessed on 10 December 2022). Clustal W2 (https://www.ebi.ac.uk/Tools/msa/clustalw2/, accessed on 15 December 2022) and MEGA 5 [71] were used for sequence analysis of NF-YA TFs. PlantPAN3.0 (http://plantpan.itps.ncku.edu.tw, accessed on 20 September 2023) was used to identify cis elements in the promoter.

### 4.6. Dual Luciferase Reporter Assay

Maize protoplasts and transient expression were prepared according to Sheen’s method [72] and used for the ZmNF-YA1 promoter-binding analysis. The promoter regions of the candidate target genes were ligated with the LUC of the Dual-LUC vector pGreenII-0800 and transformed into maize mesophyll protoplasts isolated from W22 or *nf-ya1*, or co-transformed with pAN580-PUbi1::*ZmNF-YA1*. Promoter activity is expressed as the relative activity of firefly LUC versus Renilla LUC. The effect of ZmNF-YA1 on the expression of potential target genes under heat stress was investigated. Approximately 2000 bp promoter sequences from *bZIP100* (Zm00001eb373300), *JIH5* (Zm00001eb211100), *ASN3* (Zm00001eb013430), *HUP26* (Zm00001eb406490), G*RF1*, Zm00001eb080570), and *NIP2c* (Zm00001eb372380) were amplified and inserted into the Dual-LUC vector to drive the expression of *LUC*. Primers used for vector recombination are listed in Appendix A.

### 4.7. Statistic Analysis

All data are presented as the mean values of at least three independent sets of experiments. The data are reported as the mean ± SD. For gene qRT-PCR and RNAseq analysis, samples from three biological replicates were collected at different time points during heat stress or different development stages and for subsequent analysis. Statistical analysis was performed in R. ANOVA and multiple comparisons (Tukey’s HSD) were used to compare the difference between both genotypes and conditions. For physiological parameter measurements, and at least three biological replicates were used. Student’s *t*-test was used to compare the overexpression lines, mutant to their control, respectively, and ** denotes statistical significance with *p* < 0.01, and * denotes statistical significance with *p* < 0.05.

## 5. Conclusions

The NF-YA gene family comprises a class of conserved transcription factors that are key regulatory factors in plant development and abiotic stress responses. Our previous result showed that ZmNF-YA1 could interact with the ZmNF-YB16-YC17 heterodimer to exert its biological function. To be a positive regulator of osmotic stress and to work together with ZmNF-YB16-YC17, we were interested in the role of ZmNF-YA1 in the heat stress response of maize and whether it could be a candidate for genetic engineering breeding. In this paper, the morphology and growth of maize under moderate heat stress conditions were evaluated using *ZmNF-YA1* transgenic lines and mutants. *ZmNF-YA1* was found to be a positive regulator of heat tolerance. Higher MDA content and ROS accumulation were observed in the mutant, followed by WT and overexpression lines after heat stress treatment. Transcriptome analysis showed that the loss function of *ZmNF-YA1* led to broad biological changes. One of the processes regulated by ZmNF-YA1 that plays important role in heat stress responses is protein refolding. The contribution of ZmNF-YA1 to protein stabilization, as well as the regulation of ABA, ROS, and heat/temperature signaling, may be the major reason why *ZmNF-YA1* overexpression enhanced heat tolerance and the mutant showed a heat-sensitive phenotype. ZmNF-YA1 or its ZmNF-YA1-YB16-YC17 heterotrimer could be a hub shared by multiple abiotic stress responses, particularly heat and drought. Taken together, these results suggest that ZmNF-YA1 is an ideal genetic manipulation target for improving maize thermotolerance and drought tolerance.

## Figures and Tables

**Figure 1 ijms-25-06275-f001:**
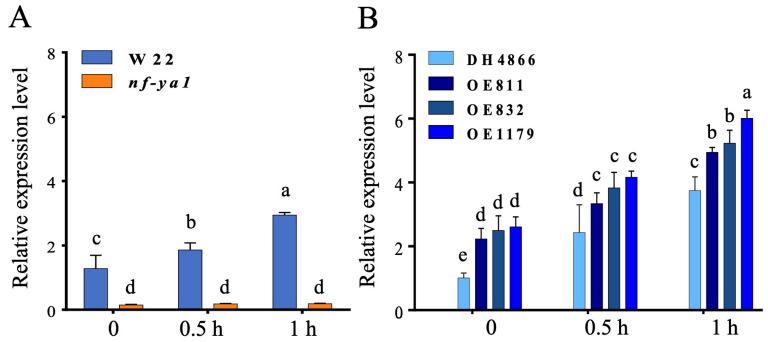
(**A**,**B**) *ZmNF-YA1* expression levels in the leaves of *ZmNF-YA1* OE and *nf-ya1* and their corresponding wild-type controls. The value of *ZmNF-YA1* expression level in WT (DH4866 and W22) before heat stress was set to 1-fold. At 39 °C, moderate heat stress was used for the heat stress treatment. *nf-ya1*, (uniformMU number: UFMu-06468), W22, maize inbred line W22, wild-type control of *Mu* insertion lines. DH4866, transgenic donor inbred line DH4866; OE811, OE832, and OE1179 are different independent *ZmNF-YA1* transgenic T3 lines. Transcript levels were calculated using the 2^−ΔΔCt^ method [49], with maize 18S rRNA [50] as an internal control. Values represent the mean of three replicates ± SD. Different letters denote statistical significance with *p* < 0.05 using ANOVA and Tukey’s HSD test.

**Figure 2 ijms-25-06275-f002:**
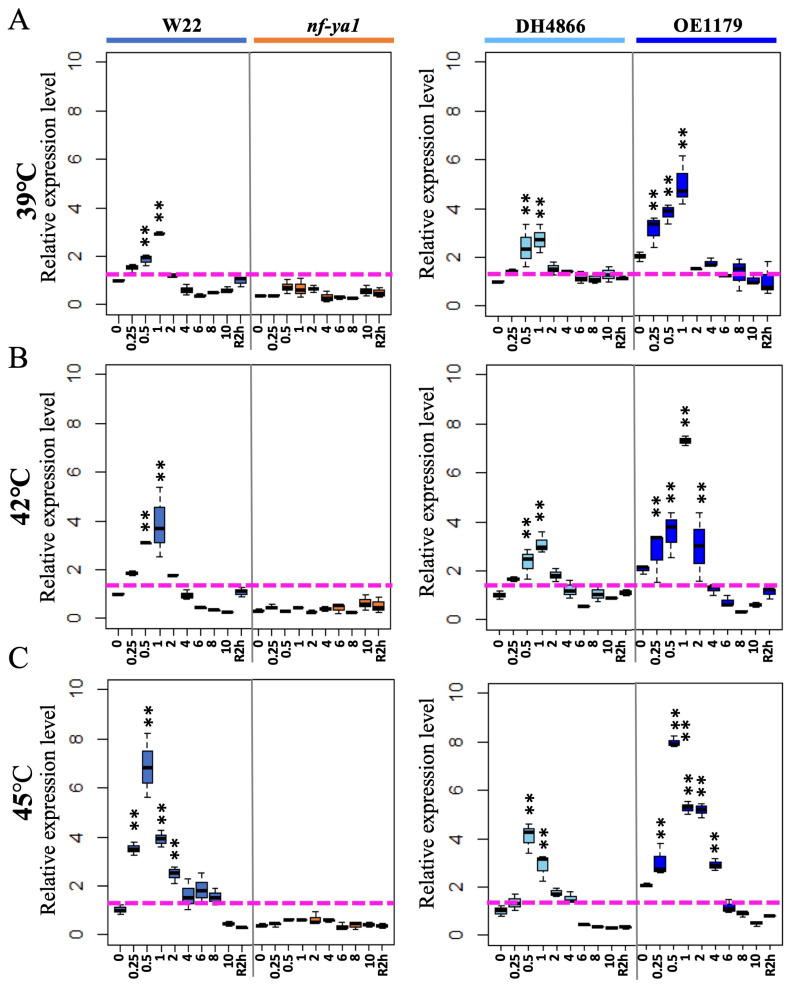
Expression profile of *ZmNF-YA1* in different lines in response to heat stress. Expression profile of *ZmNF-YA1* in leaves of *nf-ya1* mutant, W22, *ZmNF-YA1* OE, and DH4866 lines after different heat treatments: (**A**) 39 °C, moderate heat stress; (**B**) 42 °C, heat shock stress; and (**C**) 45 °C, severe heat stress. Transcript levels were calculated as described in Figure 1. Ten-day-old maize seedlings were used and samples from three biological replicates were collected at different time points during heat stress and 2 h after recovery (R2h) for qRT-PCR analysis. ** denotes statistical significance with *p* < 0.01 using a *t*-test compared with the value 0.

**Figure 3 ijms-25-06275-f003:**
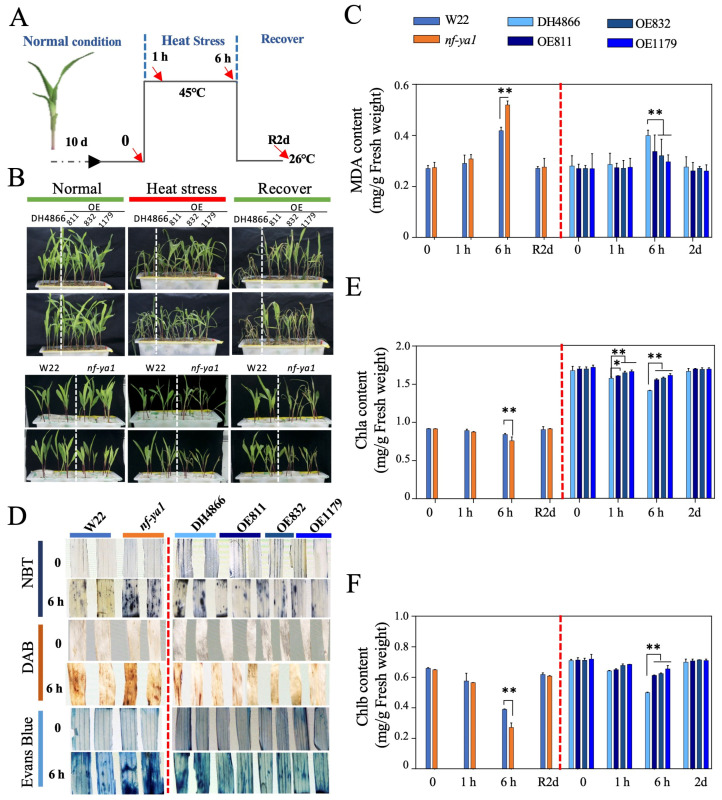
Effect of *ZmNF-YA1* on maize heat tolerance. (**A**) Schematic of the heat stress treatment of maize plants. Ten-day-old maize seedlings were subjected to 45 °C severe heat stress for 6 h and then allowed to recover. Samples from three biological replicates were collected at 0, 1, and 6 h (red arrows indicate) during heat stress and 2 days after recovery (R2d) for physiological indexes determination. (**B**) Growth of seedlings from *nf-ya1* mutant, W22, *ZmNF-YA1* OE, and DH4866 plants after heat shock treatment (45 °C, 6 h, and recovered for 2 days). (**C**) MDA content analysis; (**D**) NBT, DAB, and Evans blue staining; and chlorophyll content analysis (**E**) and (**F**) of the seedlings from *nf-ya1* mutant, W22, *ZmNF-YA1* OE, and DH4866 lines after heat stress treatment. Values represent the mean of the three replicates ± SD. ** denotes statistical significance with *p* < 0.01, and * denotes statistical significance with *p* < 0.05 using a *t*-test when compared with the value in the WT.

**Figure 4 ijms-25-06275-f004:**
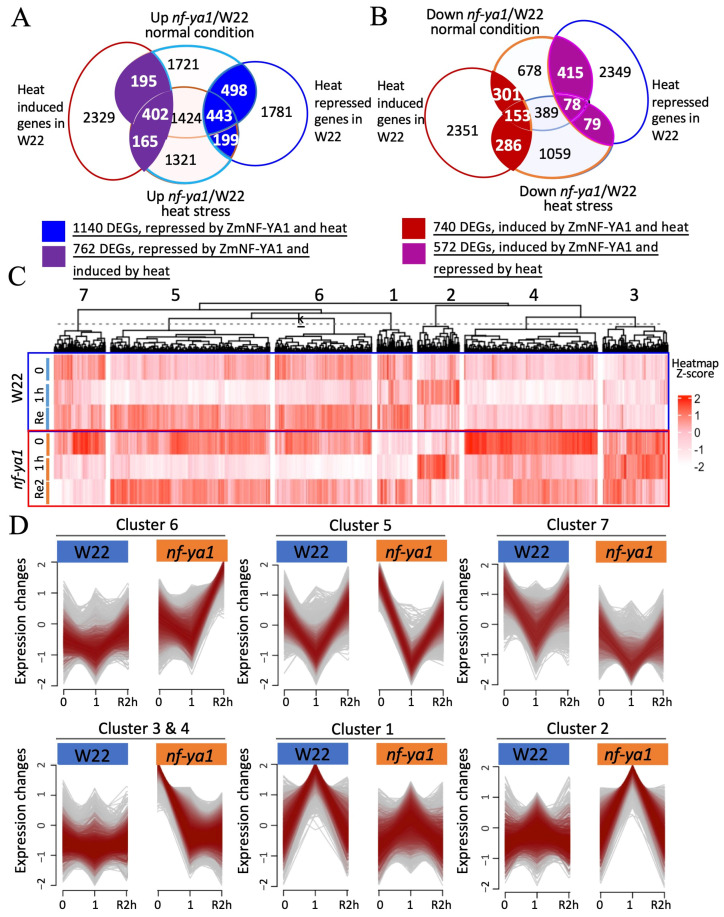
Transcriptome analysis of the *nf-ya1* mutant and W22 in response to heat stress. (**A**,**B**) Venn diagrams of DEGs in different genotypes under normal and heat stress treatments. Based on their expression patterns, DEGs affected by the interaction of ZmNF-YA1, heat stress, and ZmNF-YA1 alone were grouped into different clades. DEGs were divided into seven clusters using *k*-means clustering (**C**) and representative expression profiles in different clusters (**D**). The FDR value < 0.01 and absolute log_2_ ratio > 1 were used as cutoffs for the DEGs. The normalized count values (FPKM) were used for *k*-mean cluster analysis.

**Figure 5 ijms-25-06275-f005:**
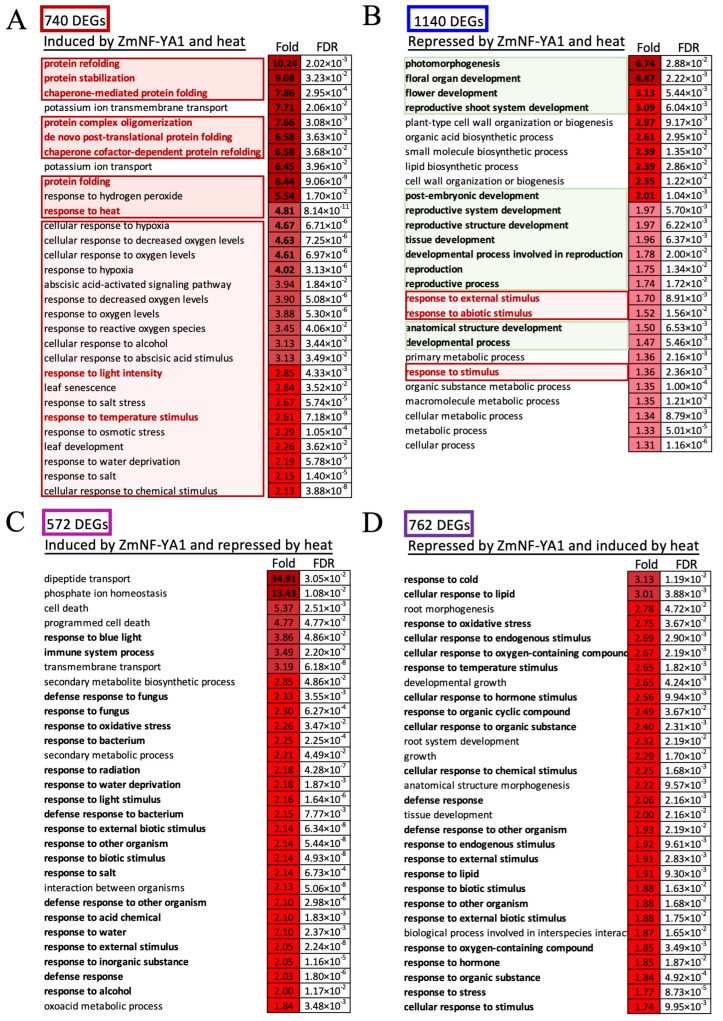
Functional analysis of the DEGs identified in the *nf-ya1* mutant and W22 in response to heat stress. (**A**) Significantly enriched GO terms of DEGs induced by ZmNF-YA1 and heat stress treatment. (**B**) Significantly enriched GO terms of DEGs repressed by ZmNF-YA1 and heat stress treatment. (**C**) Significantly enriched GO terms of DEGs induced by ZmNF-YA1 and repressed by heat stress treatment. (**D**) Significantly enriched GO terms of DEGs repressed by ZmNF-YA1 and induced by heat stress treatment. The different shades of red of the squares with different numbers indicate the enriched fold values of the Go terms. The FDR value < 0.01 and absolute log_2_ ratio > 1 were used as cutoffs for the DEGs. The top 30 or all enriched GO terms in certain groups are plotted with the enriched fold and FDR values as shown in Appendix A.

**Figure 6 ijms-25-06275-f006:**
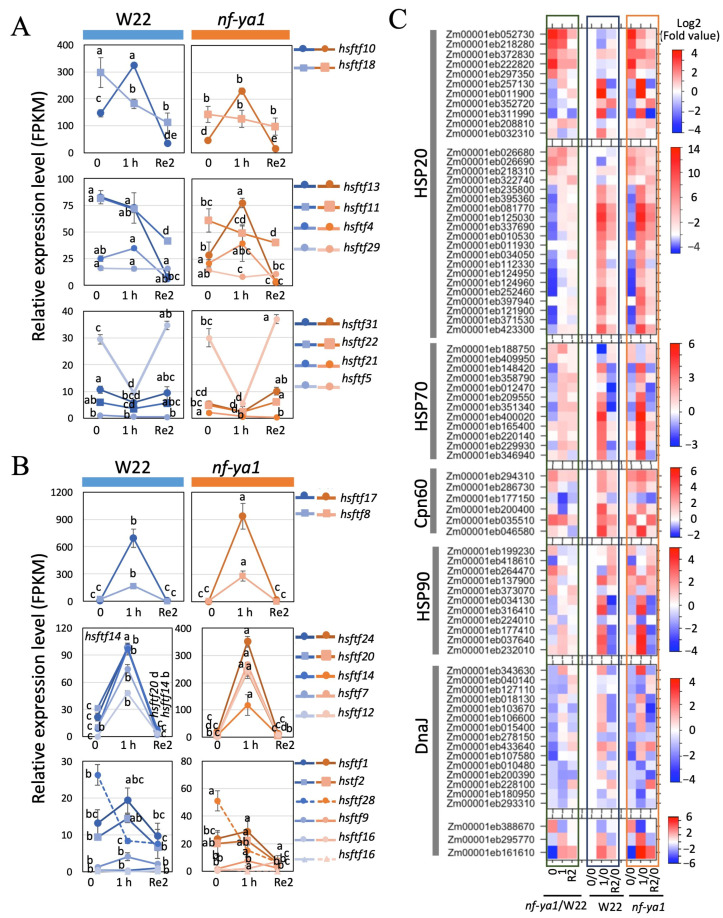
Changes in expression of heat stress response genes in *nf-ya1* and W22 in response to heat stress treatment. (**A**,**B**) line plots show the changes in the expression of heat shock transcription factors in W22 (**left panels**) and *nf-ya1*(**right panels**) in response to heat stress treatment. Values represent the mean of the three replicates ± SD; the data used and statistical analysis are shown in Appendix A. (**C**) Heat maps show the changes in the expression of different groups of HSP genes in *nf-ya1* and W22 in response to heats tress treatment. Colors represent log2 fold change in the means from triplicate samples comparing relative expression at before stress treatment (0) of W22. Different letters denote statistical significance with *p* < 0.05 using ANOVA and Tukey’s HSD test.

**Figure 7 ijms-25-06275-f007:**
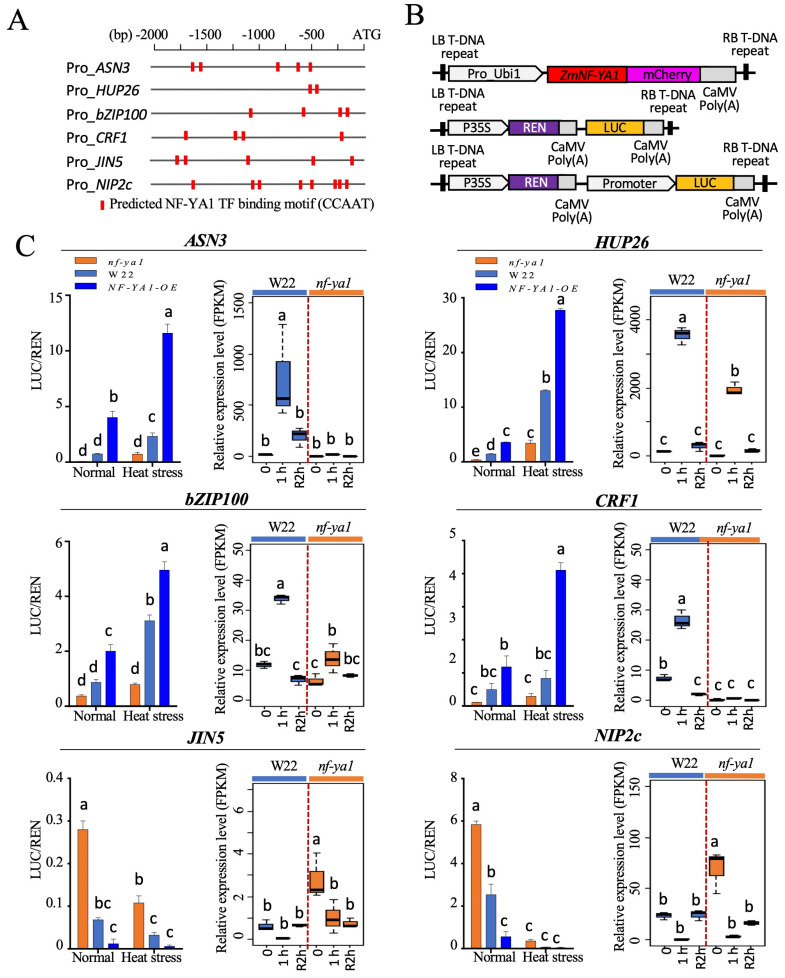
Impact of *ZmNF-YA1* on the expression of various genes in response to heat stress. (**A**) Schematic diagram of cis elements of selected promoters. (**B**) Experiment design of promoter binding assay using maize protoplast transient system. Promoters from candidate ZmNF-YA1 target genes were linked to the luciferase of the Dual-Luc vector pGreenII-0800 and transformed or co-transformed with pAN580-Pr_Ubi1::*ZmNF-YA1* in maize mesophyll protoplasts isolated from W22 or *nf-ya1*. Promoter activity is expressed as the relative activity of firefly luciferase versus Renilla luciferase. (**C**) Promoter activities (**left panel**) and expression levels (**right panel**) in cells with different *ZmNF-YA1* levels. Values represent the mean of the three replicates ± SD; different letters denote statistical significance with *p* < 0.05 using ANOVA and Tukey’s HSD test.

## Data Availability

Data is contained within the article and Appendix A. The expression data reported are available in the NCBI Gene Expression Omnibus (GEO) database under the GSE series accession number GSE268429.

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
