# Peer review of "ZmNF-YA1 Contributes to Maize Thermotolerance by Regulating Heat Shock Response"

_ijms, 2024, doi:10.3390/ijms25116275_

Round 1

Reviewer 1 Report

Comments and Suggestions for Authors

The manuscript entitled “ZmNF-YA1 contributes to maize thermotolerance by regulating heat shock response” described the positive role of ZmNF-YA1 in heat stress in maize. The author analyzed the function of ZmNF-YA1 in maize via phenotyping of zmnf-ya1 mutant and overexpression lines. The study provides new insights into understanding the mechanism of maize thermotolerance. However, this study is not well organized and written. Hence, the authors should be carefully re-check and re-organize the manuscript, and improve the English and writing.

For example,

1.      Figure 1 a, b, c and d should be deleted. It’s well detected in their previous study [45]

2.      Figure S2 can be deleted

3.      Figure S1a, the AtNF-YA3 should be added

4.      Lines 112 to 116, should be deleted

5.      Lines 24 to 26, re-organize the sentence, it’s difficult to understand.

6.      Line 17, The ZmNF-YA1 mutant should be “the nf-ya1 mutant”

7.      Lin21, delete “W22 and”

8.      Lie22, delete “at great depths”

9.      Line 23, “in nf-ya1 and W22” should be “in nf-ya1 compared to W22”

10.  Line 32, “Heat stress (HS, i.e., high temperature)” should be “Heat stress (HS)”

11.  Lines 33 to 36, re-organize the sentence

12.  Line 64, Arabidopsis should be italic

13.  Line 122, [45] should be before the dot

too many similar points, please check and correct.

Comments on the Quality of English Language

Please check manuscript and re-organize the writing and English.

Author Response

Comments and Suggestions for Authors (Reviewer 1)

The manuscript entitled “ZmNF-YA1 contributes to maize thermotolerance by regulating heat shock response” described the positive role of ZmNF-YA1 in heat stress in maize. The author analyzed the function of ZmNF-YA1 in maize via phenotyping of zmnf-ya1 mutant and overexpression lines. The study provides new insights into understanding the mechanism of maize thermotolerance. However, this study is not well organized and written. Hence, the authors should be carefully re-check and re-organize the manuscript, and improve the English and writing.

Thank you for your affirmation and encouragement of our work. And thanks so much for helping us a lot to improve this MS.

For example,

Q1: Figure 1 a, b, c and d should be deleted. It’s well detected in their previous study [45]

A1: Thank you for the suggestion. The original Figure 1 a, b, c, and d were deleted as suggested. And the corresponding figure legends were delated too.

Q2: Figure S2 can be deleted

A2: Thank you for the suggestion. The original Figure S2 was deleted as suggested. And the number of the figures followed was changed.

Q3: Figure S1a, the AtNF-YA3 should be added

A3: The original phylogenetic tree was replaced by a new phylogenetic tree, in which the numbers in Arabidopsis and rice were added, including AtNF-YA3 (See Supplemental Figure 1-4, P1). Thanks.

Q4: Lines 112 to 116, should be deleted

A4: Thank you for the suggestion. The original Lines 112-116 were deleted as suggested.

Q5: Lines 24 to 26, re-organize the sentence, it’s difficult to understand.

A5: Thank you for the suggestion. The original Lines 24-26 were revised as “ Gene Ontology (GO) enrichment analysis of the DEGs in different clades was performed to elucidate the roles of ZmNF-YA1-mediated transcriptional regulation and their contribution to maize thermotolerance. The loss function of ZmNF-YA1 led to the failure induction of DEGs in GO terms of protein refolding, protein stabilization, and GO terms for various stress responses.”.

Q6: Line 17, “The ZmNF-YA1 mutant” should be “the nf-ya1 mutant”

A6: Thanks for your correction. This “The ZmNF-YA1 mutant” was replaced by “the nf-ya1 mutant” in the revised MS.

Q7: Line 21, delete “W22 and”

A7: Thanks for your suggestion. The “W22 and” in Line 21 were deleted as suggested.

Q8: Line22, delete “at great depths”

A8: Thanks for your suggestion. The “at great depths” in Line 22 of the MS was deleted as suggested.

Q9: Line 23, “in nf-ya1 and W22” should be “in nf-ya1 compared to W22”

A9: The “in nf-ya1 and W22” in Line 23 was replaced by “in nf-ya1 compared to W22” as suggested. Thank you.

Q10: Line 32, “Heat stress (HS, i.e., high temperature)” should be “Heat stress (HS)”

A10: The “Heat stress (HS, i.e., high temperature)” in Line 32  was replaced by “Heat stress (HS)” as suggested. Thanks.

Q11: Lines 33 to 36, re-organize the sentence

A11: Thanks for your suggestion. The sentences in  Lines 33 to 36 were re-organized as follows:

P1, lines 38-40, Heat is one of the major restraints to maize plant growth, development and yield. Heat during the reproductive stages affects pollen viability, pollen growth, fertilization, and kernel development, resulting in grain yield loss[3–5].

Q12: Line 64, Arabidopsis should be italic

A13: Thanks for your correction. The Arabidopsis was italicized in the revised MS.

Q13: Line 122, [45] should be before the dot 

A13: Thanks for your correction. This was fixed in the revised MS, sorry for the carelessness.

Q14: too many similar points, please check and correct.

A14: We are sorry for the carelessness. Those points were fixed in the revised MS.

Reviewer 2 Report

Comments and Suggestions for Authors

ijms-2965873

Peer-review-v1

 The manuscript presents an interesting study in maize, a continuation of the studies revealed by https://doi.org/10.1093/plphys/kiac340, with interesting data about the ZmNF-YA1 gene versus heat stress tolerance provided. The preparation of the document lacks rigor, the presentation of some data lacks statistical support, and the discussion is poorly elaborated and little discussed with the literature. Transgenic overexpression lines are not used in validating the results observed with the knockout line, particularly with protein refolding and protein stabilization data. The study would be fantastic if everything had been done with both the knockout and overexpression lines. In brief, the data needs to be better analyzed, be better supported, and the manuscript needs profound/deep improvement in every way.

Reviewer comments:

1.      In general: Several typing errors, several editorial and formatting errors, some English errors, several poorly written sentences and phrases, some words that should be in italics are not, and some poorly chosen words in scientific terms. In particular, look to: what to write, how to write, how to put the message in the sentence, how to construct the story, how to use scientific terms correctly, and how to use information from the literature to support the message to be expressed, how to be objective and direct, among others. Citations in the middle of the sentence or phrase, but should be at the end. Erroneously, the results section is mixed of methodology, discussion, and results, while the discussion is very poorly discussed. Please, deep review.

2.      Abstract: Explain what is “W22 and nf-ya1”, “ABA, ROS” and “WT”. Not abbreviate heat shock (HS).

3.      Introduction: Not abbreviate heat shock (HS). The introduction does not focus on ZmNF-YA1, and the information about it is poorly prepared, practically uninteresting and does not attract attention (for example, but no limited, review: https://doi.org/10.3389/fpls.2023.1159955 and https://doi.org/10.1016/j.bbrc.2016.08.020; must be used deeply: https://doi.org/10.1093/plphys/kiac340). Here, "(https://climate.nasa.gov/vital-signs/global-temperature/; https://climate.nasa.gov/news/3282/)", please, format as a reference and not as a link. The last paragraph of the introduction is poorly written and completely uninteresting. Improve deeply this sentence by better introducing your work done.

4.      Results: In figure 1, information about amplicon size is truncated/unformatted. Figure 1c, what is PolyA as a terminator? All text within figures and font size must be uniform across all figures, e.g. arial 12, non-italic. For all figures with supported data with statistics shown via asterisks, it is confusing to know what is being compared. So, use different letters over each bar for statistics. Figure 2, lacks statistical support. Figures 4a and B, Venn diagrams multi-color is very confusing, in consequence of a mix of colors. Figure 4C, data are presented as rep1, rep2, and rep3 per treatment, but shouldn't these data be presented as an average of 1 treatment instead of 3 individualized repetitions? Legend of Figure 4, the 4C is duplicated. In addition to the p-value, was FDR considered for the cutoff? Figures 6a and 6c lack statistical support. Figures 6 and 7, relative expression based in FPKM was based in three libraries per treatment, so, please, provide data supported by statistical. Figure S1b lacks statistical support. How was the expression level of the endogenous ZmNF-YA1 gene versus transgenic ZmNF-YA1 differentiated in real-time RT-PCR? Figure 1D, how was the amplification of the endogenous ZmNF-YA1 gene differentiated from the ZmNF-YA1 transgene via PCR? Figure S1D is not quantitative and needs to be performed in real-time RT-PCR with normalized values. Figure 5, these data with knockout lines should be supported by including transgenic lines. Results sections 2.1 and 2.2 are poorly written:  Lots of unnecessary information and what is needed is poorly written. Which generation of mutant line and overexpression lines were used, was T4? Figure 1d is a poorly assembled figure and omits the originality of the data, and is each well 1 plant? Figure 3b, is very low quality/resolution for any visual analysis (can be a problem during editorial loading). Figure 3d, there is no quantitative data supported by statistics, therefore it is impossible to evaluate. They can be used, but additional data and figures or tables with statistical support should be provided. In the transcriptome analysis should be included at least 1 overexpression line. The analysis of DEGs was poor and limited, potentially omitting other functions and implications of the ZmNF-YA1 gene. Figure 4a, the Venn diagram, the number of genes per contrast, and diagrams are very confusing as presented. Other analyses should be performed, in addition to "Gene Ontology (GO) enrichment analysis", such as PageMan, MapMan, and KEGG pathway enrichments. All data shown in Figure 6 lacks statistical support. Some (5-10) downstream genes of ZmNF-YA1 should be selected and monitored by RT-PCR, supporting the data shown in Figure 7 and including at least 1 transgenic overexpression line.

5.      From the RNA-seq data provide additional analyses of PageMan, MapMan, and KEGG (bubble plots) enrichment analysis.

6.      In a table, RNA-seq versus RT-PCR data should be validated by Pearson correlation coefficient, using TPM or TPKM values per RNA-seq library versus (2^-deltaCt) values per library of RT-PCR.

7.      Supplemental material: Figure S1b lacks statistical support. Tables Suppl. with legend poorly written and lacks information to understand. Suppl. Table S6, includes gene ID numbers, melting temperature, and amplicon size. What is the "F+R" primer?

8.      Discussion: presents the results again and poorly discussed with support from the literature. This topic needs notable improvements.

9.      Materials and Methods: In the 4.1 Plant materials subsection, should be mentioned that both knockout mutant and overexpression lines were obtained by Yang et al. [45]. This information is repeated two times “RNA was extracted from a small sample (~0.1 g) of leaf lamina (excluding the midrib) from the middle of the first fully expanded leaf.”, please, re-write. In RT-PCR, “three biological replicates were used”, and what number of plants per replicate? For chlorophyll, MDA, and staining analysis, "leaves samples were collected", what leaf was evaluated, from top or base? And what experimental design was used? This sentence “Primers were designed using NCBI/primer-BLAST.” Should be in the RT-PCR section. 

Comments on the Quality of English Language

Should be improved.

Author Response

Comments and Suggestions for Authors (Reviewer 2)

The manuscript presents an interesting study in maize, a continuation of the studies revealed by https://doi.org/10.1093/plphys/kiac340, with interesting data about the ZmNF-YA1 gene versus heat stress tolerance provided. The preparation of the document lacks rigor, the presentation of some data lacks statistical support, and the discussion is poorly elaborated and little discussed with the literature. Transgenic overexpression lines are not used in validating the results observed with the knockout line, particularly with protein refolding and protein stabilization data. The study would be fantastic if everything had been done with both the knockout and overexpression lines. In brief, the data needs to be better analyzed, be better supported, and the manuscript needs profound/deep improvement in every way.

Thank you for your affirmation and encouragement of our work. And thanks so much for helping us a lot to improve this MS.

Reviewer comments:

Q1: In general: Several typing errors, several editorial and formatting errors, some English errors, several poorly written sentences and phrases, some words that should be in italics are not, and some poorly chosen words in scientific terms. In particular, look to: what to write, how to write, how to put the message in the sentence, how to construct the story, how to use scientific terms correctly, and how to use information from the literature to support the message to be expressed, how to be objective and direct, among others. Citations in the middle of the sentence or phrase, but should be at the end. Erroneously, the results section is mixed of methodology, discussion, and results, while the discussion is very poorly discussed. Please, deep review. 

A1: Thanks so much for helping us to improve this MS. We are sorry for the careless and mistakes. Thanks for your great suggestion on writing, data organization, and interpretation. We have made extensive modification on the original manuscript and giving a point- by-point response to the concerns. Now, the revised manuscript was resubmitted by using the "Track Changes" to check. Thanks again for your great suggestion.

Q2: Abstract: Explain what is “W22 and nf-ya1”, “ABA, ROS” and “WT”. Not abbreviate heat shock (HS).

A2: Thanks for your suggestion. The full name of W22, nf-ya1, “ABA, ROS” and “WT” were added in the revised MS the no “HS” used, thanks.

Q3: Introduction: Not abbreviate heat shock (HS). The introduction does not focus on ZmNF-YA1, and the information about it is poorly prepared, practically uninteresting and does not attract attention (for example, but no limited, review: https://doi.org/10.3389/fpls.2023.1159955 and https://doi.org/10.1016/j.bbrc.2016.08.020; must be used deeply: https://doi.org/10.1093/plphys/kiac340). Here, "(https://climate.nasa.gov/vital-signs/global-temperature/; https://climate.nasa.gov/news/3282/)", please, format as a reference and not as a link. The last paragraph of the introduction is poorly written and completely uninteresting. Improve deeply this sentence by better introducing your work done.

A3: Thanks for your suggestion.

HS was replaced by using “heat stress” in the revised MS, thanks.

The published papers suggested by the reviewers were added to the revised MS and deeply described as suggested.

The format of the two links was revised as suggested.

Q4: See the answers for Q4.1-Q4.12, thanks.

Q4.1: In figure 1, information about amplicon size is truncated/unformatted. Figure 1c, what is PolyA as a terminator?

A4.1: For the T-DNA region of the construct (Original Figure 1c), we have a ZmNF-YA1 gene driven by RD 29A promoter and terminated by Tnos (3’-utr), and the herbicide-resistant gene bar driven by P35S and terminated with CaMV35S ploly A. We are sorry for the mistake. The original Figure 1a-d was deleted in the revised MS as suggested by another reviewer because all the info was described in our published paper cited. Instead, the brief descriptions of the T-DNA region of the expression construct were added in the revised MS.

Q4.2: All text within figures and font size must be uniform across all figures, e.g. arial 12, non-italic.

A4.2: We are sorry for the carelessness. Those points were fixed in the revised MS.

Q4.3: For all figures with supported data with statistics shown via asterisks, it is confusing to know what is being compared. So, use different letters over each bar for statistics. Figure 2, lacks statistical support.

A4.3: Thanks for your correction. The ANOVA and letter labels were used to show the statistical difference of each group in original Figures 1e and f (Figures 1 a and b in the revised MS). The statistical analysis by using student t-test was used in Figures 2 and 3,  and the connecting lines were added in the revised Figures to show the comparison and what is compared.

Q4.4: Figures 4a and B, Venn diagrams multi-color is very confusing, in consequence of a mix of colors. Figure 4C, data are presented as rep1, rep2, and rep3 per treatment, but shouldn't these data be presented as an average of 1 treatment instead of 3 individualized repetitions? Legend of Figure 4, the 4C is duplicated. In addition to the p-value, was FDR considered for the cutoff?

A4.4: Thanks for your suggestion. The colors in Figures 4a and b in the Venn diagrams were revised to avoid confusion. Figure 4c was replotted using the mean of three reps as suggested. Legend of Figure 4, the 4c is fixed. Thanks for your correction. FDR was used to cutoff, I am sorry for the mistake. Those were fixed in the revised MS.

Q4.5: Figures 6a and 6c lack statistical support. Figures 6 and 7, relative expression based in FPKM was based in three libraries per treatment, so, please, provide data supported by statistical. Figure S1b lacks statistical support.

A4.5: Thanks for your suggestion. The ANOVA and letter labels were added in the revised MS as suggested in the revised MS. The letter labels were added in Figure 7c. For Figure 6, a table with statistical analysis was added in the supplemental table S6 (supporting data of Figure 6) because an expression changes the trends of multiple genes in one panel and is difficult to label. The corresponding figure legends were fixed too. Thanks again.

Q4.6: How was the expression level of the endogenous ZmNF-YA1 gene versus transgenic ZmNF-YA1 differentiated in real-time RT-PCR? Figure 1D, how was the amplification of the endogenous ZmNF-YA1 gene differentiated from the ZmNF-YA1 transgene via PCR? 

A4.6: Thanks for your suggestion. I am not sure if my understanding is right for this point. The original Figure 1d is the gel image of PCR analysis of the ZmNF-YA1 overexpressing T3 transgenic maize by using the primers for the herbicide-resistant marker bar gene of the construct. The amplification is for the exogenous bar gene not for ZmNF-YA1. Herbicide screening and PCR of the bar gene helped us to select the transgenic lines and then qRT-PCR was used to select the lines with transgene significantly overexpressed. We have tried the primer sets with the combination of primers target to 3’-utr, however, this construct, is not feasible because of the high sequence similarity of the genes in ZmNF-YA1 clade.

Q4.7: Figure S1D is not quantitative and needs to be performed in real-time RT-PCR with normalized values. 

A4.7: Thanks for your suggestion. The real-time RT-PCR result of the tissue expression profiles was added, and the figure legends were fixed.

Q4.8: Figure 5, these data with knockout lines should be supported by including transgenic lines. 

A4.8: Thanks for your suggestion. A qRT-PCR for the key HSFs in Figures 5 and 6 in different lines with 1 overexpression line and their WT was added in the revised MS as Figure S7. Thanks again.

Q4.9: Results sections 2.1 and 2.2 are poorly written:  Lots of unnecessary information and what is needed is poorly written. Which generation of mutant line and overexpression lines were used, was T4? 

A4.9: Thanks for your suggestion. This section was revised according to the suggestion. The plant material (mutant line and overexpression lines) and other information were added. see details in Sections 2.1 and 2.2.

Q4.10: Figure 1d is a poorly assembled figure and omits the originality of the data, and is each well 1 plant? 

A4.10: Thanks for your suggestion. We are sorry for the mistake. Each well comes from one individual plant and four plants for each independent transgenic line. This gel showed all the selected lines are homozygous. And the original Figure 1a-1d was deleted as suggested by another reviewer because the molecular identification was well described in our previous paper cited. Thanks.

Q4.10: Figure 3b, is very low quality/resolution for any visual analysis (can be a problem during editorial loading). Figure 3d, there is no quantitative data supported by statistics, therefore it is impossible to evaluate. They can be used, but additional data and figures or tables with statistical support should be provided.

A4.10: Thanks for your suggestion. We tried to use a new version with high resolution (Figure 3b) to make it clear to see. Hope it works well.  For Figure 3d, we tried to qualification, however, it is difficult and not Inaccurate in ImageJ. Not sure if there any method works. If any choices,  please let me know, will try to make it work.

Q4.11: In the transcriptome analysis should be included at least 1 overexpression line. The analysis of DEGs was poor and limited, potentially omitting other functions and implications of the ZmNF-YA1 gene. 

A4.11: Great suggestion. This is our preliminary experiment design of the transcriptome analysis. We have a time-course with different heat stress because of the stress-induced RD29A promoter was used to drive the expression of transgene. Both the mutant and OE lines have their inbred line control. It is complex and difficult to interpret the result because two maize inbred lines in the assay, and the transgene is a stress-induced manner. The correlation of the ZmNF-YA1 levels to the downstream targets is not good or easy to find as that using the constitutive promoter. more data such as Chip-seq or DAPseq are needed for that group analysis. A suggestion from an expert suggested not to put too many things in one paper, that is impossible. That is the major reason why the paper is the current structure.  Another reason not to go into too detail about the RNAseq analysis, especially the DGEs between genotypes is some of the results already in our previous paper. So slightly difficult to draw the story without repeating. That is why we start from the DEGs under the regulation of both heat and ZmNF-YA1. In the revised MS, the KO and pathway and the highly correlated biological processes related to ZmNF-YA1 were added as Figures S4, S5 and S6. Instead of using the RNAseq for the OE lines, qRT-PCR was used to validate the key targets such as the DEGs in Figures 5, 6 and 7 in all the lines including the OE lines and their WT DH4866. Thanks so much for your help in improving our MS and the suggestion of the data interpretation. Thanks again.

Figure 4a, the Venn diagram, the number of genes per contrast, and diagrams are very confusing as presented. 

Thanks for your suggestion. The colors in the vein diagram were fixed as suggested. Thanks.

Q4.12: Other analyses should be performed, in addition to "Gene Ontology (GO) enrichment analysis", such as PageMan, MapMan, and KEGG pathway enrichments. From the RNA-seq data provide additional analyses of PageMan, MapMan, and KEGG (bubble plots) enrichment analysis.

Thanks for your suggestion. GO, molecular function, and pathway analysis were added and discussed in the text in the revised MS as suggested (Figure S4, S5, and S6, bubble plot, KEGG ).

All data shown in Figure 6 lacks statistical support. Some (5-10) downstream genes of ZmNF-YA1 should be selected and monitored by RT-PCR, supporting the data shown in Figure 7 and including at least 1 transgenic overexpression line.

The ANOVA and letter labels were added in the revised MS as suggested in the revised MS. The letter labels were added in Figure 7c. For Figure 6, a table with statistical analysis was added in the supplemental table S6 (supporting data of Figure 6) because an expression changes the trends of multiple genes in one panel and is difficult to label. The corresponding figure legends were fixed too. Thanks again.

10 downstream genes of ZmNF-YA1 should be selected and monitored by RT-PCR was added in Figure S3, with the Pearson correlation coefficients, Thanks for your suggestion.

For Figure 7, a qRT-PCR in different lines with at least 1 overexpression line and their WT was added in the revised MS as Figure S8. Thanks again.

Q5: In a table, RNA-seq versus RT-PCR data should be validated by Pearson correlation coefficient, using TPM or TPKM values per RNA-seq library versus (2^-deltaCt) values per library of RT-PCR.

A5: Thanks for your suggestion. The Pearson correlation coefficient of the RNAseq (FPKM) vs. qRT-PCR result was added (Figure S3).

Q6: Supplemental material: Figure S1b lacks statistical support. Tables Suppl. with legend poorly written and lacks information to understand. Suppl. Table S6, includes gene ID numbers, melting temperature, and amplicon size. What is the "F+R" primer?

A6: Thanks for your suggestion. This section was revised according to the suggestion. The statistical result was added for Figure S1b.  The information in Tables S1 and S6 were added as suggested.

Q7: Discussion: presents the results again and poorly discussed with support from the literature. This topic needs notable improvements.

A7: Thanks for your suggestion. The Discussion section was reorganized and the comparison to the literature was compared in the revised MS. See the revised Discussion section.

Q8: Materials and Methods: In the 4.1 Plant materials subsection, should be mentioned that both knockout mutant and overexpression lines were obtained by Yang et al. [45].

Thanks for your suggestion. This section was re-write according to the suggestion with both the mutant and overexpression lines obtained by Yang et al. see details in Section 4.1.

This information is repeated two times “RNA was extracted from a small sample (~0.1 g) of leaf lamina (excluding the midrib) from the middle of the first fully expanded leaf.”, please, re-write.

“RNA was extracted from a small sample (~0.1 g) of leaf lamina (excluding the midrib) from the middle of the first fully expanded leaf” at the end of section 4.2 was deleted. Thanks for your correction.

In RT-PCR, “three biological replicates were used”, and what number of plants per replicate? For chlorophyll, MDA, and staining analysis, "leaves samples were collected", what leaf was evaluated, from top or base? And what experimental design was used? This sentence “Primers were designed using NCBI/primer-BLAST.” Should be in the RT-PCR section. 

Thanks for your suggestion. This section was re-write according to the suggestion. For all the parameters including the qRT-PCR and physiological index, each biological replicate contained the leaf fragments from 3-4 plants grown under the same conditions. The 1st fully expanded leaf from the top was used, and the plants were at 3-leaf stage. The sentence “Primers were designed using NCBI/primer-BLAST.” was moved to the RT-PCR section.  See section 4.2, Thanks.

Reviewer 3 Report

Comments and Suggestions for Authors

Dear Authors,

In the manuscript „ZmNF-YA1 contributes to maize thermotolerance by regulating heat shock response“ it has been investigated whether ZmNF-YA1 confers heat stress tolerance in Zea mays. ZmNF-YA1 overexpression enhanced heat tolerance, and the mutant showed a heat-sensitive phenotype. ZmNF-469 YA1 is benefit to protein refolding and stabilization when subjected to HS.

The manuscript is with high scientific relevance and provide novel information. The Authors have done huge amount of work, but in my opinion there is a lot of information in the manuscript and it will be difficult for the readers to follow up the text. Results and Discusion sections must be corrected and improved.

I have some suggestions and comments which will be useful to improve the structure and vision of the manuscript:

1.      When you mention plant type for first time use the scientific name first and than point the common name – line 11 – Zea mays (maize); line 79 – „in rice“, should be corrected to „in Oryza sativa (rice)“

2.      The name of the genes should be italicized, and the name of the proteins should not be italicized; please verify this throughout the manuscript

3.      Line 102 – full stop is missing after „ZmBT2[42]“.

4.      Try not to comment and discues on the results. Comments and suggestions should be made in the Discussion – line 121-122; line 127-129; line 137-138; line 145-149; line 221-223; line 280-290; line 306-309; line 311-318; line 323-329; line 337-340; line 345-351; line 377-379

5. The text under the Figures should be shortened.

Author Response

Comments and Suggestions for Authors (Reviewer 3)

Dear Authors,

In the manuscript „ZmNF-YA1 contributes to maize thermotolerance by regulating heat shock response“ it has been investigated whether ZmNF-YA1 confers heat stress tolerance in Zea mays. ZmNF-YA1 overexpression enhanced heat tolerance, and the mutant showed a heat-sensitive phenotypeZmNF-469 YA1 is benefit to protein refolding and stabilization when subjected to HS.

The manuscript is with high scientific relevance and provide novel information. The Authors have done huge amount of work, but in my opinion there is a lot of information in the manuscript and it will be difficult for the readers to follow up the text. Results and Discusion sections must be corrected and improved.

Thank you for your affirmation and encouragement of our work. And thanks so much for helping us a lot to improve this MS.

I have some suggestions and comments which will be useful to improve the structure and vision of the manuscript:

Q1: When you mention plant type for first time use the scientific name first and than point the common name – line 11 – Zea mays (maize); line 79 – „in rice“, should be corrected to „in Oryza sativa (rice)“

A1: Sorry for the careless. Those were fixed in the revised MS.

Q2: The name of the genes should be italicized, and the name of the proteins should not be italicized; please verify this throughout the manuscript

A2: Sorry for the careless. Those were fixed in the revised MS, see the revised MS.

Q3: Line 102 – full stop is missing after „ZmBT2[42]“.

A3: Thanks for your careful examination. This was fixed in the revised MS as suggested.

Q4: Try not to comment and discues on the results. Comments and suggestions should be made in the Discussion – line 121-122; line 127-129; line 137-138; line 145-149; line 221-223; line 280-290; line 306-309; line 311-318; line 323-329; line 337-340; line 345-351; line 377-379

A4: Thanks for your suggestion. The corresponding sections mentioned above were deleted from the result section and moved into the discussion section. See the revised MS for details. Thanks.

Q5: The text under the Figures should be shortened.

A5: Thanks for your suggestion. The figure legends under the figures were shortened as suggested.

Reviewer 4 Report

Comments and Suggestions for Authors

The manuscript is focused on exploring the role of a Nuclear factor Y family member, namely ZmNF-YA1, in promoting thermotolerance in maize. This was accomplished by using both ZmNF-YA1 mutants and overexpression lines to investigate the functional significance of this gene in maize. The work presented in the manuscript is well-structured, scientifically sound, and designed in a manner that is sufficient for readers to clearly understand the methodology and experimental design utilized in the study. The introduction is well-written and provides a clear overview of the research problem being addressed. Overall, the study is interesting and falls squarely within the scope of the IJMS journal. However, a few issues must be addressed before the manuscript can be accepted for publication.

1.      The gene name should be italicized.

2.      The figure 1b, 000 should be 1000.

3.      The statistical analysis section should also be included in the material and method section.

4.      The conclusion of the study should be included in the manuscript.

Author Response

Comments and Suggestions for Authors (Reviewer 4)

The manuscript is focused on exploring the role of a Nuclear factor Y family member, namely ZmNF-YA1, in promoting thermotolerance in maize. This was accomplished by using both ZmNF-YA1 mutants and overexpression lines to investigate the functional significance of this gene in maize. The work presented in the manuscript is well-structured, scientifically sound, and designed in a manner that is sufficient for readers to clearly understand the methodology and experimental design utilized in the study. The introduction is well-written and provides a clear overview of the research problem being addressed. Overall, the study is interesting and falls squarely within the scope of the IJMS journal. However, a few issues must be addressed before the manuscript can be accepted for publication.

Q1: The gene name should be italicized.

A1: Sorry for the careless. Those were fixed in the revised MS.

Q2: The figure 1b, 000 should be 1000.

A2: Sorry for the careless. Those were fixed. And the original Figure 1a-1d was deleted as suggested by another reviewer because the molecular identification was well described in our previous paper cited. Thanks.

Q3: The statistical analysis section should also be included in the material and method section.

A3: Thanks for your suggestion. A section for statistical analysis was added in the M&M. Thanks again.

Q4: The conclusion of the study should be included in the manuscript.

A4: Thanks for your suggestion. the conclusion of the study was added in the revised MS.

Round 2

Reviewer 2 Report

Comments and Suggestions for Authors

ijms-2965873-peer-review-v2

Dear authors, thank you for the opportunity to read again your study. All suggestions and corrections were properly addressed and explained, and the current version of this manuscript was significantly improved. A few minor points are still needed to improve.

Minor comments:

1.       Several punctual typing and writing errors should be corrected after a careful review.

2.       The legends of Figures S2 to S8 are poorly elaborated. Please, improve the description.

3.       Figure S6a and 6b, green boxes are up or down-regulated genes? red boxes are up or down-regulated genes? Were these data used and discussed?

4.       Suppl. Table S6, provide also gene ID.

5.       Figure S1d, remove the image of the electrophoresis gel since it is uninformative. Figure S1e provides statistics support based on “different letters” comparing all against all.

6.       Figures S4a and S5a, were selected protein-folding chaperones, but other pathways were up-represented compared to this.

7.       Figure 3d, there is no quantitative data supported by statistics, therefore it is impossible to evaluate. They can be used, but additional data and figures or tables with statistical support should be provided. Values obtained from ImageJ, each treatment with at least three biological replicates, some technical replicates per biological replicate, should be evaluated quantitatively and supported by statistics.

8.       Overall, the resolution of all figures is fully poor in the PDF generated to review (this must be a problem when editing the PDF file).

9.       I do not recommend the abbreviation of heat stress response (HSR) and heat shock factors (HSFs). What is the functional difference between HSF and HSP? Excessive abbreviations contribute to confusion.

10.   Here, “Our previous result showed that ZmNF-YA1 could interact with the ZmNF-YB16-YC17 heterodimer to exert their biological function. The evolutionary tree was conducted using ZmNF-YA1 and the members in this subgroup”, provide the citation/reference.

11.   Previous data about AtNF-YA3 and GmNF-YA3 should be used in the discussion section, which are putative orthologs of ZmNF-YA1.

12.   In the Discussion section, should be addressed and discussed in detail the differential and similar ZmNF-YA1 functional roles in maize response to drought (previous study) versus heat (this study) stresses, since both studies characterized the nf-ya1 homozygous mutant lines.

13.   Here, “Atrd29A (RD29A)” change to “AtRD29A” and “(Prd29A driven) change to “driven by RD29 promoter”.

14.   Here, “model analysis using Dr. Tom”, provides the citation/reference.

15.   Here, “different genotypes and HSRs”, the HSRs term can be inadequate, change to “heat stress conditions”.

16.   Here, for example, “The HSR DEGs”, is confusing this term, please, improve.

17.   Figures 6A and 6B, please, provide the statistical support within this figure, in addition or instead of Supplemental Table 6.

18.   What cis-elements can the NF-YA1 protein bind to? Only to CCAAT motif? Provide citation/reference. Only 6 DEGs contain this motif? However, ok about the selected 6 targets for validation. Provide a short explanation in the 2.7 subsection.

19.   The JIH5 and NIP2c promoters contain several CCAAT motifs but were not activated by NF-YA1, also supported by RNA-seq data.  Explain briefly about them in the 2.7 subsection. NF-YA1 acts as a repressor to JIH5 and NIP2c?

20.   The discussion section should be improved by discussing results obtained with NF-YA1 orthologous in other plant species. The 3.1 subsection is poorly discussed with literature data.

21.   The Discussion section poorly explores the expression regulation of multiple HSFs by ZmNF-YA1. Add a new subsection or should be addressed in the 3.2 subsection.

22.   The 3.3 subsection did not critically compare the results obtained from ZmNF-YA1 and drought versus ZmNF-YA1 and heat, as well as, additional data from the literature.

23.   In the 4.7 subsection, please, explain which data were evaluated in a particular manner. Throughout the manuscript, when pertinent, the number of biological replicates per treatment, the number of technical replicates per sample, and the number of plants per biological replicate must be duly informed.

Comments on the Quality of English Language

No comments.

Author Response

Comments and Suggestions for Authors (Reviewer 2, round2)

Dear authors, thank you for the opportunity to read again your study. All suggestions and corrections were properly addressed and explained, and the current version of this manuscript was significantly improved. A few minor points are still needed to improve.

Thank you for your affirmation and encouragement of our work. And thanks so much for helping us a lot to improve this MS.

Minor comments:

  1. Several punctual typing and writing errors should be corrected after a careful review.

A1: Thanks so much for helping us to improve this MS. We are sorry for the careless and mistakes. We have checked the MS carefully. Now, the revised manuscript was resubmitted by using the "Track Changes" to check. Thanks again for your great suggestion.

  1. The legends of Figures S2 to S8 are poorly elaborated. Please, improve the description.

A2: Thanks for your suggestion. This section was revised according to the suggestion. See details in legends of Figures S2 to S8.

  1. Figure S6a and 6b, green boxes are up or down-regulated genes? red boxes are up or down-regulated genes? Were these data used and discussed?

A3: Thanks for your carefully examination. The green boxes indicate the downregulated genes in the comparasion and red boxes indicate the upregulated genes. The use of the data was disccussed in the revised MS, and the legends Figures S6 was revised as suggested.

  1. Suppl. Table S6, provide also gene ID.

A4: Thanks for your suggestion. A column of gene ID was added in Suppl. Table S6.

  1. Figure S1d, remove the image of the electrophoresis gel since it is uninformative. Figure S1e provides statistics support based on “different letters” comparing all against all.

A5: Thanks for your suggestion. This section was revised according to the suggestion.

  1. Figures S4a and S5a, were selected protein-folding chaperones, but other pathways were up-represented compared to this. 

A6: Thanks for your suggestion. The enriched pathways were listed in the figure S5b and S6b, with more than a dozen pathways significantly enriched. We are focused on the potential transcriptional regulation of ZmNF-YA1, especially abiotic stress sensing, signal transduction, and some hubs' work upstream of the general stress responses. The pathways on the top were related to metabolism such as Glycolysis, starch and sucrose metabolism, etc. The alternation or acclimation of those pathways may be due to the altered cell status. In other words, it maybe an indirect effect, and not specific to the heat stress response.  To focus on the transcriptional regulation of ZmNF-YA1 mediated in response to heat stress, protein processing (unfolded protein response in ER and Cytoplasmic) was mainly discussed in this paper. Some of the other key DEGs and mediated stress responses or development were interesting and more experimental evidence is needed except for the expression changes. We hope that part of the work goes well. Thanks.

  1. Figure 3d, there is no quantitative data supported by statistics, therefore it is impossible to evaluate. They can be used, but additional data and figures or tables with statistical support should be provided. Values obtained from ImageJ, each treatment with at least three biological replicates, some technical replicates per biological replicate, should be evaluated quantitatively and supported by statistics.

A7: Thanks for your suggestion. Quantitative data and statistical analysis were added in Figure S2 as suggested.

  1. Overall, the resolution of all figures is fully poor in the PDF generated to review (this must be a problem when editing the PDF file). 

A8: Thanks for your suggestion. We tried other methods to generate the PDF. If needed, we will ask the assistant editor for help to make it clear to read.

  1. I do not recommend the abbreviation of heat stress response (HSR) and heat shock factors (HSFs). What is the functional difference between HSF and HSP? Excessive abbreviations contribute to confusion.

A9: Thanks for your suggestion. The abbrevations of HSR and HSF was revised according to the suggestion. And the relationship of HSF and HSP was added in the revised MS.

  1. Here, “Our previous result showed that ZmNF-YA1 could interact with the ZmNF-YB16-YC17 heterodimer to exert their biological function. The evolutionary tree was conducted using ZmNF-YA1 and the members in this subgroup”, provide the citation/reference.

A10: Thanks for your suggestion.  The citation was added in the revised MS.

  1. Previous data about AtNF-YA3 and GmNF-YA3 should be used in the discussion section, which are putative orthologs of ZmNF-YA1.

A11: Thanks for your suggestion. The function of  AtNF-YA3, GmNF-YA3, and orthologs in rice was added in the discussion section of the revised MS. See details Section 3.1 in the revised MS.

  1. In the Discussion section, should be addressed and discussed in detail the differential and similar ZmNF-YA1 functional roles in maize response to drought (previous study) versus heat (this study) stresses, since both studies characterized the nf-ya1 homozygous mutant lines.

A12: Thanks for your suggestion. A paragraph of the comparison of ZmNF-YA1-mediated transcriptional regulation in response to heat and drought stress was added as suggested. See details in the revised MS. Thanks.

  1. Here, “Atrd29A (RD29A)” change to “AtRD29A” and “(Prd29A driven) change to “driven by RD29 promoter”.

A13: Thanks for your suggestion. This section was revised according to the suggestion.

  1. Here, “model analysis using Dr. Tom”, provides the citation/reference.

A14: Thanks for your suggestion.  The citation was added in the revised MS.

  1. Here, “different genotypes and HSRs”, the HSRs term can be inadequate, change to “heat stress conditions”.

A15: Thanks for your suggestion. The abbrevations of HSR and HSF was revised according to the suggestion.

  1. Here, for example, “The HSR DEGs”, is confusing this term, please, improve. 

A16: Thanks for your suggestion. The abbrevations of HSR and the The “HSR DEGs” was revised according to the suggestion.

  1. Figures 6A and 6B, please, provide the statistical support within this figure, in addition or instead of Supplemental Table 6. 

A17: Thanks for your suggestion. The statistical analysis was added in Figure 6a and b in the revised MS.

  1. What cis-elements can the NF-YA1 protein bind to? Only to CCAAT motif? Provide citation/reference. Only 6 DEGs contain this motif? However, ok about the selected 6 targets for validation. Provide a short explanation in the 2.7 subsection. 

A18: Thanks for your suggestion. The promoter analysis we are founced on was added in the revised MS. The “CCAAT box” was core and conical motifs in the promoter of NF-YA regulated target genes. Some of the scombination such as the combinnation to ERSE motif as ERSEII were reported in certain genes like BIP3 in A.th, however, the universal or common NF-YA bignding motifs are still largrly unknow except for the  “CCAAT box”. An explanation was added in the revised MS as suggested. Thanks.

  1. The JIH5 and NIP2c promoters contain several CCAAT motifs but were not activated by NF-YA1, also supported by RNA-seq data.  Explain briefly about them in the 2.7 subsection. NF-YA1 acts as a repressor to JIH5 and NIP2c?

A19: Thanks for your suggestion. A short explanation and conclusion of this section was added in the revised MS as suggested. Thanks.

  1. The discussion section should be improved by discussing results obtained with NF-YA1 orthologous in other plant species. The 3.1 subsection is poorly discussed with literature data.

A20: Thanks for your suggestion. The function of  AtNF-YA3, GmNF-YA3, and orthologs in rice was added in the discussion section of the revised MS. See details in Section 3.1 in the revised MS.

  1. The Discussion section poorly explores the expression regulation of multiple HSFs by ZmNF-YA1. Add a new subsection or should be addressed in the 3.2 subsection.

A21: Thanks for your suggestion. A paragraph on the expression changes of multiple HSFs and the potential regulation mechanisms was added in section 3.2 of the revised MS. See details in the discussion section of the revised MS.

  1. The 3.3 subsection did not critically compare the results obtained from ZmNF-YA1 and drought versus ZmNF-YA1 and heat, as well as, additional data from the literature.

A22: Thanks for your suggestion. A paragraph of the comparison of ZmNF-YA1-mediated transcriptional regulation in response to heat and drought stress was added as suggested. See details in the discussion section of the revised MS.Thanks.

  1. In the 4.7 subsection, please, explain which data were evaluated in a particular manner. Throughout the manuscript, when pertinent, the number of biological replicates per treatment, the number of technical replicates per sample, and the number of plants per biological replicate must be duly informed.

A23: Thanks for your suggestion. We fixed the 4.7 subsection as suggested.

Reviewer 3 Report

Comments and Suggestions for Authors

Dear Authors,

I am satisfied with the changes and corrections made to the text. In my opinion, the manuscript is ready for publication in its current form.

Author Response

Thank you for your affirmation and encouragement of our work. And thanks so much for helping us a lot to improve this MS.

Round 3

Reviewer 2 Report

Comments and Suggestions for Authors

ijms-2965873

Dear authors, thank you for the opportunity to read again your study. A nice paper with powerful results. There are still some minor writing/typing errors that should be reviewed now or during the proof review. 

Recommendation: Accept

Comments on the Quality of English Language

No comments.